# Scaling White-Box Transformers for Vision

**Jinrui Yang**[*1]  **Xianhang Li**[*1]  **Druv Pai**[2]

**Yuyin Zhou**[1]  **Yi Ma**[2]  **Yaodong Yu**[†2]  **Cihang Xie**[†1]

[*]equal technique contribution, [†]equal advising

[1]UC Santa Cruz     [2]UC Berkeley

## Abstract

CRATE, a white-box transformer architecture designed to learn compressed and sparse representations, offers an intriguing alternative to standard vision transformers (ViTs) due to its inherent mathematical interpretability. Despite extensive investigations into the scaling behaviors of language and vision transformers, the scalability of CRATE remains an open question which this paper aims to address. Specifically, we propose CRATE-$\alpha$, featuring strategic yet minimal modifications to the sparse coding block in the CRATE architecture design, and a light training recipe designed to improve the scalability of CRATE. Through extensive experiments, we demonstrate that CRATE-$\alpha$ can effectively scale with larger model sizes and datasets. For example, our CRATE-$\alpha$-B substantially outperforms the prior best CRATE-B model accuracy on ImageNet classification by 3.7%, achieving an accuracy of 83.2%. Meanwhile, when scaling further, our CRATE-$\alpha$-L obtains an ImageNet classification accuracy of 85.1%. More notably, these model performance improvements are achieved while preserving, and potentially even enhancing the interpretability of learned CRATE models, as we demonstrate through showing that the learned token representations of increasingly larger trained CRATE-$\alpha$ models yield increasingly higher-quality unsupervised object segmentation of images. The project page is https://rayjryang.github.io/CRATE-alpha/.

## 1 Introduction

Over the past several years, the Transformer architecture [42] has dominated deep representation learning for natural language processing (NLP), image processing, and visual computing [8, 2, 9, 5, 12]. However, the design of the Transformer architecture and its many variants remains largely empirical and lacks a rigorous mathematical interpretation. This has largely hindered the development of new Transformer variants with improved efficiency or interpretability. The recent white-box Transformer model CRATE [46] addresses this gap by deriving a simplified Transformer block via unrolled optimization on the so-called *sparse rate reduction* representation learning objective.

More specifically, layers of the white-box CRATE architecture are mathematically derived and fully explainable as unrolled gradient descent-like iterations for optimizing the sparse rate reduction. The self-attention blocks of CRATE explicitly conduct compression via denoising features against learned low-dimensional subspaces, and the MLP block is replaced by an incremental sparsification (via ISTA [1, 11]) of the features. As shown in previous work [47], besides mathematical interpretability, the learned CRATE models and features also have much better semantic interpretability than conventional transformers, i.e., visualizing features of an image naturally forms a zero-shot image segmentation of that image, even when the model is only trained on classification.

Scaling model size is widely regarded as a pathway to improved performance and emergent properties [44, 40, 41, 14]. Until now, the deployment of CRATE has been limited to relatively modest scales. The most extensive model described to date is the base model size encompasses 77.6M parameters

38th Conference on Neural Information Processing Systems (NeurIPS 2024).

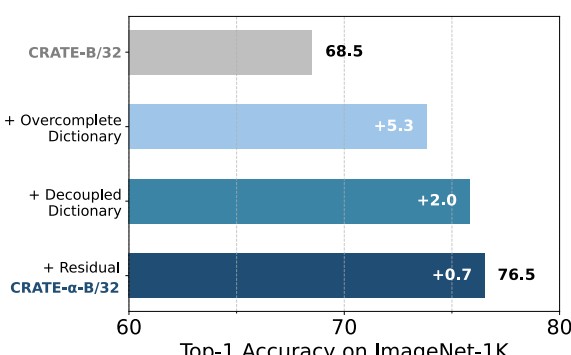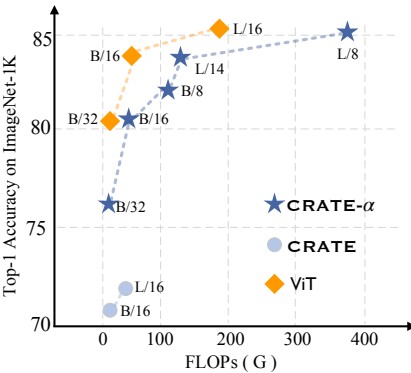

Figure 1: *(Left)* We demonstrate how modifications to the components enhance the performance of the CRATE model. The four models are trained using the same setup: first pre-trained on ImageNet-21K and then fine-tuned on ImageNet-1K. Details are provided in Section 3. *(Right)*. We compare the FLOPs and accuracy on ImageNet-1K of our methods with ViT [9] and CRATE [46]. The values of CRATE-$\alpha$ model correspond to those presented in Table 1. A more detailed comparison between CRATE-$\alpha$ and ViT is included in Appendix A.2.

(CRATE-Large) [46]. This contrasts sharply with standard Vision Transformers (ViTs [9]), which have been effectively scaled to a much larger model size, namely 22B parameters [5].

To this end, this paper provides the first exploration of training CRATE at different scales for vision, i.e., Tiny, Small, Base, Large, Huge. Detailed model specifications are given in Table 7 of Appendix A.1. To achieve effective scaling, we make two key changes. First, we identify the vanilla ISTA block within CRATE as a limiting factor that hinders further scaling. To overcome this, we significantly expand the channels, decouple the association matrix, and add a residual connection, resulting in a new model variant — CRATE-$\alpha$. It is worth noting that this architecture change still preserves the mathematical interpretability of the model. Second, we propose an improved training recipe, inspired by previous work [38, 46, 39], for better coping the training with our new CRATE-$\alpha$ architecture.

We provide extensive experiments supporting the effective scaling of our CRATE-$\alpha$ models. For example, we scale the CRATE-$\alpha$ model from Base to Large size for supervised image classification on ImageNet-21K [6], achieving *85.1% top-1 accuracy on ImageNet-1K* at the Large model size. We further scale the model size from Large to Huge, utilizing vision-language pre-training with contrastive learning on DataComp1B [10], and achieve *a zero-shot top-1 accuracy of 72.3% on ImageNet-1K* at the Huge model size.[1] These results demonstrate the strong scalability of the CRATE-$\alpha$ model, shedding light on scaling up mathematically interpretable models for future work.

The main contributions of this paper are threefold:

1. We design three strategic yet minimal modifications for the CRATE model architecture to unleash its potential. In Figure 1, we reproduce the results of the CRATE model within our training setup, initially pre-training on ImageNet-21K classification and subsequently fine-tuning on ImageNet-1K classification. Compared to the vanilla CRATE model that achieves 68.5% top-1 classification accuracy on ImageNet-1K, our CRATE-$\alpha$-B/32 model significantly improves the vanilla CRATE model by 8%, which clearly demonstrates the benefits of the three modifications to the existing CRATE model. Moreover, following the settings of the best CRATE model and changing the image patch size from 32 to 8, our CRATE-$\alpha$-B model attains a top-1 accuracy of 83.2% on ImageNet-1K, exceeding the previous best CRATE model's score of 79.5% by a significant margin of 3.7%.

2. Through extensive experiments, we show that one can effectively scale CRATE-$\alpha$ via model size and data simultaneously. In contrast, when increasing the CRATE model from Base to Large model size, there is a marginal improvement on top-1 classification accuracy (+0.5%, from 70.8% to 71.3%) on ImageNet-1K, indicating diminished returns [46]. Furthermore, by scaling the training dataset, we achieved a substantial 1.9% improvement in top-1 classification accuracy on ImageNet-1K, increasing from 83.2% to 85.1% when going from CRATE-$\alpha$ Base to Large.

3. We further successfully scale CRATE-$\alpha$ model from Large to Huge by leveraging vision-language pre-training on DataComp1B. Compared to the Large model, the Huge model (CRATE-$\alpha$-H) achieves a zero-shot top-1 classification accuracy of 72.3% on ImageNet-1K, marking a significant

---

[1]Model configurations are detailed in Table 7 (in Appendix A.1).

scaling gain of 2.5% over the Large model. These results indicate that the CRATE architecture has the potential to serve as an effective backbone for vision-language foundation models.

### Related Work

**White-box Transformers.** [46, 45] argued that the quality of a learned representation can be assessed through a unified objective function called the *sparse rate reduction*. Based on this framework, [46, 45] developed a family of transformer-like deep network architectures, named CRATE, which are mathematically fully interpretable. CRATE models has been demonstrably effective on various tasks, including vision self-supervised learning and language modeling [26, 45]. Nevertheless, it remains unclear whether CRATE can scale as effectively as widely used black-box transformers. Previous work [46] suggests that scaling the vanilla CRATE model can be notably challenging.

**Scaling ViT.** ViT [9] represents the initial successful applications of Transformers to the image domain on a large scale. Many works [12, 31, 33, 32, 5, 37, 21, 22, 29, 18, 49] have deeply explored various ways of scaling ViTs in terms of model size and data size. From the perspective of self-supervision, MAE [12] provides a scalable approach to effectively training a ViT-Huge model using only ImageNet-1K. Following the idea of MAE, [31] further scales both model parameters to billions and data size to billions of images. Additionally, CLIP was the first to successfully scale ViT on a larger data scale (i.e., 400M) using natural language supervision. Based on CLIP, [32, 33] further scale the model size to 18 billion parameters, named EVA-CLIP-18B, achieving consistent performance improvements with the scaling of ViT model size. From the perspective of supervised learning, [49, 5] present a comprehensive analysis of the empirical scaling laws for vision transformers on image classification tasks, sharing some similar conclusions with [15]. [49] suggests that the performance-compute frontier for ViT models, given sufficient training data, tends to follow a saturating power law. More recently, [5] scales up ViT to 22 billion parameters. Scaling up different model architectures is non-trivial. [37, 21, 22] have made many efforts to effectively scale up different architectures. In this paper, due to the lack of study on the scalability of white-box models, we explore key architectural modifications to effectively scale up white-box transformers in the image domain.

## 2  Background and Preliminaries

In this section, we present the background on white-box transformers proposed in [46], including representation learning objectives, unrolled optimization, and model architecture. We first introduce the notation that will be used in the later presentation.

**Notation.** We use notation and problem setup following Yu et al. [46]. We use the matrix-valued random variable $\boldsymbol{X} = [\boldsymbol{x}_1, \ldots, \boldsymbol{x}_N] \in \mathbb{R}^{D \times N}$ to represent the data, where each $\boldsymbol{x}_i \in \mathbb{R}^D$ is a "token", such that each data point is a realization of $\boldsymbol{X}$. For instance, $\boldsymbol{X}$ can represent a collection of image patches for an image, and $\boldsymbol{x}_i$ is the $i$-th image patch. We use $f \in \mathcal{F} \colon \mathbb{R}^{D \times N} \to \mathbb{R}^{d \times N}$ to denote the mapping induced by the transformer, and we let $\boldsymbol{Z} = f(\boldsymbol{X}) = [\boldsymbol{z}_1, \ldots, \boldsymbol{z}_N] \in \mathbb{R}^{d \times N}$ denote the features for input data $\boldsymbol{X}$. Specifically, $\boldsymbol{z}_i \in \mathbb{R}^d$ denotes the feature of the $i$-th input token $\boldsymbol{x}_i$. The transformer $f$ consists of multiple, say $L$, layers, and so can be written as $f = f^L \circ \cdots \circ f^1 \circ f^{\mathrm{pre}}$, where $f^\ell \colon \mathbb{R}^{d \times N} \to \mathbb{R}^{d \times N}$ denotes the $\ell$-th layer of the transformer, and the pre-processing layer is denoted by $f^{\mathrm{pre}} = \mathbb{R}^{D \times N} \to \mathbb{R}^{d \times N}$. The input to the $\ell$-th layer $f^\ell$ of the transformer is denoted by $\boldsymbol{Z}^\ell = [\boldsymbol{z}_1^\ell, \ldots, \boldsymbol{z}_N^\ell] \in \mathbb{R}^{d \times N}$, so that $f^\ell \colon \boldsymbol{Z}^\ell \mapsto \boldsymbol{Z}^{\ell+1}$. In particular, $\boldsymbol{Z}^1 = f^{\mathrm{pre}}(\boldsymbol{X}) \in \mathbb{R}^{d \times N}$ denotes the output of the pre-processing layer and the input to the first layer.

### 2.1  Sparse Rate Reduction

Following the framework proposed in [45], we posit that the goal of representation learning is to learn a feature mapping or *representation* $f \in \mathcal{F} \colon \mathbb{R}^{D \times N} \to \mathbb{R}^{d \times N}$ that transforms the input data $\boldsymbol{X}$ (which may have a nonlinear, multi-modal, and otherwise complicated distribution) into *structured and compact* features $\boldsymbol{Z}$, such that the token features lie on a union of low-dimensional subspaces, say with orthonormal bases $\boldsymbol{U}_{[K]} = (\boldsymbol{U}_k)_{k \in [K]} \in (\mathbb{R}^{d \times p})^K$. [46] proposes the *Sparse **R**ate **R**eduction* (SRR) *objective* to measure the goodness of such a learned representation:

$$\max_{f \in \mathcal{F}} \mathbb{E}_{\boldsymbol{Z}=f(\boldsymbol{X})}\left[L_{\mathrm{srr}}(\boldsymbol{Z})\right] = \min_{f \in \mathcal{F}} \mathbb{E}_{\boldsymbol{Z}=f(\boldsymbol{X})}\left[R^c(\boldsymbol{Z} \mid \boldsymbol{U}_{[K]}) - R(\boldsymbol{Z} \mid \boldsymbol{U}_{[K]}) + \lambda\|\boldsymbol{Z}\|_1\right], \quad (1)$$

where $\boldsymbol{Z} = f(\boldsymbol{X})$ denotes the token representation, $\|\boldsymbol{Z}\|_1$ denotes the $\ell^1$ norm, and $R(\boldsymbol{Z})$, $R^c(\boldsymbol{Z} \mid \boldsymbol{U}_{[K]})$ are (estimates for) *rate distortions* [4, 7], defined as:

$$R(\boldsymbol{Z}) \doteq \frac{1}{2} \log \det \left( \boldsymbol{I} + \frac{d}{N\epsilon^2} \boldsymbol{Z}^\top \boldsymbol{Z} \right), \qquad R^c(\boldsymbol{Z} \mid \boldsymbol{U}_{[K]}) \doteq \sum_{k=1}^{K} R(\boldsymbol{U}_k^\top \boldsymbol{Z}). \tag{2}$$

In particular, $R^c(\boldsymbol{Z} \mid \boldsymbol{U}_{[K]})$ (resp. $R(\boldsymbol{Z})$) provide closed-form estimates for the number of bits required to encode the sample $\boldsymbol{Z}$ up to precision $\epsilon > 0$, conditioned (resp. unconditioned) on the samples being drawn from the subspaces with bases $\boldsymbol{U}_{[K]}$. Minimizing the term $R^c$ improves the compression of the features $\boldsymbol{Z}$ against the posited model, and maximizing the term $R$ promotes non-collapsed features. The remaining term $\lambda\|\boldsymbol{Z}\|_1$ promotes sparse features. Refer to [45] for more details about the desiderata and objective of representation learning via the rate reduction approach.

## 2.2 CRATE: Coding RATE Transformer

**Unrolled optimization.** To optimize the learning objective and learn compact and structured representation, one approach is unrolled optimization [11, 36]: each layer of the deep network implements an iteration of an optimization algorithm on the learning objective. For example, one can design the layer $f^\ell$ such that the forward pass is equivalent to a proximal gradient descent step for optimizing learning objective $L(\boldsymbol{Z})$, i.e., $\boldsymbol{Z}^{\ell+1} = f^\ell(\boldsymbol{Z}^\ell) = \texttt{Prox}[\boldsymbol{Z}^\ell - \eta \cdot \nabla_{\boldsymbol{Z}} L(\boldsymbol{Z}^\ell)]$. Here we use $\eta$ to denote the step size and $\texttt{Prox}[\cdot]$ to denote the proximal operator [27].

**One layer of the CRATE model.** We now present the design of each layer of the white-box transformer architecture – Coding RATE Transformer (CRATE) – proposed in [46]. Each layer of CRATE contains two blocks: the compression block and the sparsification block. These correspond to a two-step alternating optimization procedure for optimizing the sparse rate reduction objective (1). Specifically, the $\ell$-th layer of CRATE is defined as

$$\boldsymbol{Z}^{\ell+1} = f^\ell(\boldsymbol{Z}^\ell) = \texttt{ISTA}(\boldsymbol{Z}^{\ell+1/2} \mid \boldsymbol{D}^\ell), \quad \text{where} \quad \boldsymbol{Z}^{\ell+1/2} = \boldsymbol{Z}^\ell + \texttt{MSSA}(\boldsymbol{Z}^\ell). \tag{3}$$

**Compression block (MSSA).** The compression block in CRATE, called **M**ulti-head **S**ubspace **S**elf-**A**ttention block (MSSA), is derived for compressing the token set $\boldsymbol{Z} = [\boldsymbol{z}_1, \dots, \boldsymbol{z}_N]$ by optimizing the compression term $R^c$ (defined Eq. (1)), i.e.,

$$\boldsymbol{Z}^{\ell+1/2} = \boldsymbol{Z}^\ell + \texttt{MSSA}(\boldsymbol{Z}^\ell \mid \boldsymbol{U}_{[K]}^\ell) \approx \boldsymbol{Z}^\ell - \kappa \nabla_{\boldsymbol{Z}} R^c(\boldsymbol{Z}^\ell \mid \boldsymbol{U}_{[K]}^\ell), \tag{4}$$

where $\boldsymbol{U}_{[K]}^\ell$ denotes the (local) signal model at layer $\ell$, and the MSSA operator is defined as

$$\texttt{MSSA}(\boldsymbol{Z} \mid \boldsymbol{U}_{[K]}) = \frac{\kappa p}{N\epsilon^2} [\boldsymbol{U}_1 \cdots \boldsymbol{U}_K] \begin{bmatrix} (\boldsymbol{U}_1^\top \boldsymbol{Z}) \operatorname{softmax}((\boldsymbol{U}_1^\top \boldsymbol{Z})^\top (\boldsymbol{U}_1^\top \boldsymbol{Z})) \\ \vdots \\ (\boldsymbol{U}_K^\top \boldsymbol{Z}) \operatorname{softmax}((\boldsymbol{U}_K^\top \boldsymbol{Z})^\top (\boldsymbol{U}_K^\top \boldsymbol{Z})) \end{bmatrix}. \tag{5}$$

Compared with the commonly used attention block in transformer [42], where the $k$-th attention head is defined as $(\boldsymbol{V}_k^\top \boldsymbol{Z}) \operatorname{softmax}((\boldsymbol{Q}_k^\top \boldsymbol{Z})^\top (\boldsymbol{K}_k^\top \boldsymbol{Z}))$, MSSA uses only one matrix to obtain the query, key, and value matrices in the attention: that is, $\boldsymbol{U}_k = \boldsymbol{Q}_k = \boldsymbol{K}_k = \boldsymbol{V}_k$.

**Sparse coding block (ISTA).** The Iterative Shrinkage-Thresholding Algorithm (ISTA) block is designed to optimize the sparsity term and the global coding rate term, $\lambda\|\boldsymbol{Z}\|_0 - R(\boldsymbol{Z} \mid \boldsymbol{U}_{[K]})$ in (1). [46] shows that an optimization strategy for these terms posits a (complete) incoherent dictionary $\boldsymbol{D}^\ell \in \mathbb{R}^{d \times d}$ and takes a proximal gradient descent step towards solving the associated LASSO problem $\arg\min_{\boldsymbol{Z} \geq \boldsymbol{0}} [\frac{1}{2}\|\boldsymbol{Z}^{\ell+1/2} - \boldsymbol{D}^\ell \boldsymbol{Z}\|_2^2 + \lambda\|\boldsymbol{Z}\|_1]$, obtaining the iteration

$$\boldsymbol{Z}^{\ell+1} = \texttt{ISTA}(\boldsymbol{Z}^{\ell+1/2} \mid \boldsymbol{D}^\ell) = \operatorname{ReLU}(\boldsymbol{Z}^{\ell+1/2} + \eta (\boldsymbol{D}^\ell)^\top (\boldsymbol{Z}^{\ell+1/2} - \boldsymbol{D}^\ell \boldsymbol{Z}^{\ell+1/2}) - \eta\lambda\boldsymbol{1}). \tag{6}$$

In particular, the ISTA block sparsifies the intermediate iterates $\boldsymbol{Z}^{\ell+1/2}$ w.r.t. $\boldsymbol{D}^\ell$ to obtain $\boldsymbol{Z}^{\ell+1}$.

## 3 CRATE-$\alpha$ Model

In this section, we present the CRATE-$\alpha$ architecture, which is a variant of CRATE [46]. As shown in Fig. 1 (*Right*), there is a significant performance gap between the white-box transformer CRATE-B/16

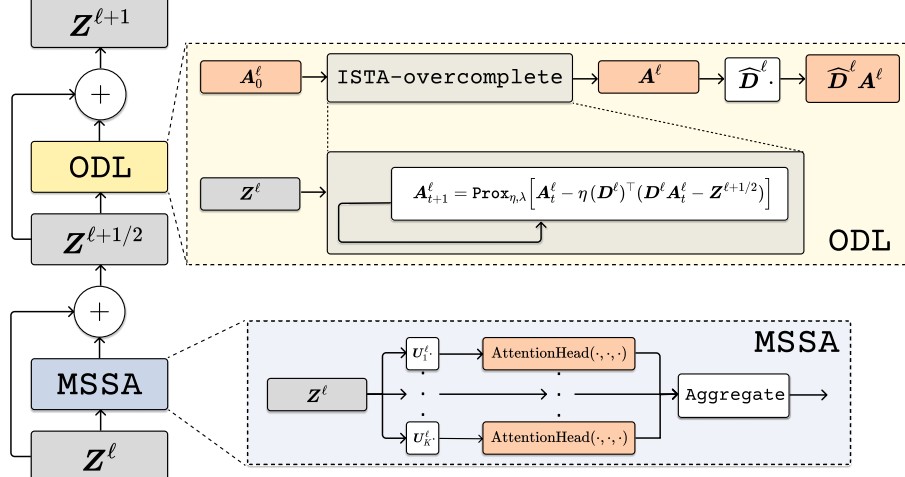

Figure 2: One layer of the CRATE-$\alpha$ model architecture. MSSA (**M**ulti-head **S**ubspace **S**elf-**A**ttention, defined in (5)) represents the compression block, and ODL (**O**vercomplete **D**ictionary **L**earning, defined in (12)) represents the sparse coding block. A more detailed illustration of the modifications is provided in Fig. 6 in the Appendix .

(70.8%) and the vision transformer ViT-B/16 (84.0%) [9]. One possible reason is that the ISTA block applies a complete dictionary $\boldsymbol{D} \in \mathbb{R}^{d \times d}$, which may limit its expressiveness. In contrast, the MLP block in the transformer[2] applies two linear transformations $\boldsymbol{W}_1, \boldsymbol{W}_2 \in \mathbb{R}^{d \times 4d}$, leading to the MLP block having 8 times more parameters than the ISTA block.

Since the ISTA block in CRATE applies a single incremental step to optimize the sparsity objective, applying an orthogonal dictionary can make it ineffective in sparsifying the token representations. Previous work [28] has theoretically demonstrated that overcomplete dictionary learning enjoys a favorable optimization landscape. In this work, we use an overcomplete dictionary in the sparse coding block to promote sparsity in the features. Specifically, instead of using a complete dictionary $\boldsymbol{D}^\ell \in \mathbb{R}^{d \times d}$, we use an overcomplete dictionary $\boldsymbol{D}^\ell \in \mathbb{R}^{d \times (Cd)}$, where $C > 1$ (a positive integer) is the overcompleteness parameter. Furthermore, we explore two additional modifications to the sparse coding block that lead to improved performance for CRATE. We now describe the three variants of the sparse coding block that we use in this paper.

**Modification #1: Overparameterized sparse coding block.** For the output of the $\ell$-th CRATE attention block $\boldsymbol{Z}^{\ell+1/2}$, we propose to sparsify the token representations with respect to an overcomplete dictionary $\boldsymbol{D}^\ell \in \mathbb{R}^{d \times (Cd)}$ by optimizing the following LASSO problem,

$$\boldsymbol{A}^\ell \approx \underset{\boldsymbol{A} \geq \boldsymbol{0}}{\arg\min} \left[ \frac{1}{2} \|\boldsymbol{Z}^{\ell+1/2} - \boldsymbol{D}^\ell \boldsymbol{A}\|_2^2 + \lambda \|\boldsymbol{A}\|_1 \right]. \tag{7}$$

To approximately solve (7), we apply two steps of proximal gradient descent, i.e.,

$$\boldsymbol{A}_0^\ell = \boldsymbol{0}, \qquad \boldsymbol{A}_1^\ell = \text{Prox}_{\eta,\lambda}\left[\boldsymbol{A}_0^\ell; \boldsymbol{D}^\ell, \boldsymbol{Z}^{\ell+1/2}\right], \qquad \boldsymbol{A}_2^\ell = \text{Prox}_{\eta,\lambda}\left[\boldsymbol{A}_1^\ell; \boldsymbol{D}^\ell, \boldsymbol{Z}^{\ell+1/2}\right], \tag{8}$$

where Prox is the proximal operator of the above non-negative LASSO problem (7) and defined as

$$\text{Prox}_{\eta,\lambda}[\boldsymbol{A}; \boldsymbol{D}, \boldsymbol{Z}] = \text{ReLU}(\boldsymbol{A} - \eta \boldsymbol{D}^\top (\boldsymbol{D}\boldsymbol{A} - \boldsymbol{Z}) - \eta\lambda\boldsymbol{1}). \tag{9}$$

The output of the sparse coding block is defined as

$$\boldsymbol{Z}^{\ell+1} = \boldsymbol{D}^\ell \boldsymbol{A}^\ell, \quad \text{where} \quad \boldsymbol{A}^\ell = \boldsymbol{A}_2^\ell \doteq \text{ISTA-OC}(\boldsymbol{Z}^{\ell+1/2} \mid \boldsymbol{D}^\ell). \tag{10}$$

Namely, $\boldsymbol{A}^\ell$ is a sparse representation of $\boldsymbol{Z}^{\ell+1/2}$ with respect to the overcomplete dictionary $\boldsymbol{D}^\ell$. The original CRATE ISTA tries to learn a complete dictionary $\boldsymbol{D} \in \mathbb{R}^{d \times d}$ to transform and sparsify the features $\boldsymbol{Z}$. By leveraging more atoms than the ambient dimension, the overcomplete dictionary $\boldsymbol{D} \in \mathbb{R}^{d \times (Cd)}$ can provide a redundant yet expressive codebook to identify the salient sparse

---

[2]The MLP block is defined as $\boldsymbol{Z}^{\ell+1} = \boldsymbol{Z}^\ell + \boldsymbol{W}_2\sigma(\boldsymbol{W}_1^\top \boldsymbol{Z}^{\ell+1/2})$, where $\sigma$ is the nonlinear activation function and $\boldsymbol{Z}^{\ell+1/2}$ denotes the output of the attention block.

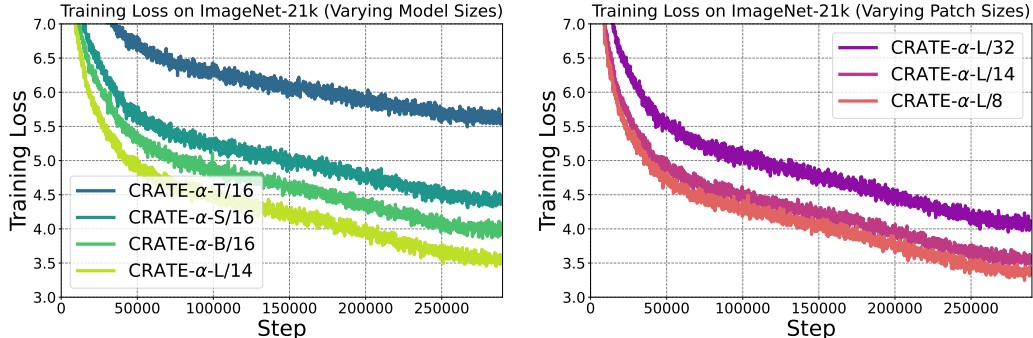

Figure 3: Training loss curves of CRATE-$\alpha$ on ImageNet-21K. (*Left*) Comparing training loss curves across CRATE-$\alpha$ with different model sizes. (*Right*) Comparing training loss curves across CRATE-$\alpha$-Large with different patch sizes.

structures underlying $\boldsymbol{Z}$. As shown in Fig. 1, the overcomplete dictionary design leads to $5.3\%$ improvement compared to the vanilla CRATE model.

**Modification #2: Decoupled dictionary.** We propose to apply a decoupled dictionary $\widehat{\boldsymbol{D}}^\ell$ in the last step (defined in (10)) of the sparse coding block, $\boldsymbol{Z}^{\ell+1} = \widehat{\boldsymbol{D}}^\ell \boldsymbol{A}^\ell$, where $\widehat{\boldsymbol{D}}^\ell \in \mathbb{R}^{d \times (Cd)}$ is a different dictionary compared to $\boldsymbol{D}^\ell$. By introducing the decoupled dictionary, we further improve the model performance by $2.0\%$, as shown in Fig. 1. We denote this mapping from $\boldsymbol{Z}^{\ell+1/2}$ to $\boldsymbol{Z}^{\ell+1}$ as the **O**vercomplete **D**ictionary **L**earning block (ODL), defined as follows:

$$\texttt{ODL}(\boldsymbol{Z}^{\ell+1/2} \mid \boldsymbol{D}^\ell, \widehat{\boldsymbol{D}}^\ell) \doteq \widehat{\boldsymbol{D}}^\ell \cdot \texttt{ISTA-OC}(\boldsymbol{Z}^{\ell+1/2} \mid \boldsymbol{D}^\ell) = \widehat{\boldsymbol{D}}^\ell \boldsymbol{A}^\ell. \tag{11}$$

**Modification #3: Residual connection.** Based on the previous two modifications, we further add a residual connection, obtaining the following modified sparse coding block:

$$\boldsymbol{Z}^{\ell+1} = \boldsymbol{Z}^{\ell+1/2} + \texttt{ODL}(\boldsymbol{Z}^{\ell+1/2} \mid \boldsymbol{D}^\ell, \widehat{\boldsymbol{D}}^\ell). \tag{12}$$

An intuitive interpretation of this modified sparse coding block is as follows: instead of directly sparsifying the feature representations $\boldsymbol{Z}$, we first identify the potential sparse patterns present in $\boldsymbol{Z}$ by encoding it over a learned dictionary. Subsequently, we incrementally refine $\boldsymbol{Z}$ by exploiting the sparse codes obtained from the previous encoding step. From Fig. 1, we find that the residual connection leads to a $0.7\%$ improvement.

To summarize, to effectively scale white-box transformers, we implement three modifications to the vanilla white-box CRATE model proposed in [46]. Specifically, in our CRATE-$\alpha$ model, we introduce a decoupling mechanism, quadruple the dimension of the dictionary ($4\times$), and incorporate a residual connection in the sparse coding block.

## 4 Experiments

**Overall.** The experimental section consists of three parts: (1) **Scaling study:** We thoroughly investigate the scaling behaviors of CRATE-$\alpha$ from Base to Large size and ultimately to Huge size. (2) **Downstream applications:** To further verify the broader benefits of scaling the CRATE-$\alpha$ model, we conduct additional experiments on real-world downstream tasks and present preliminary exploration results of CRATE-$\alpha$ on language tasks. (3) **Interpretability:** In addition to scalability, we study the interpretability of CRATE-$\alpha$ across different model sizes.

### 4.1 Dataset and Evaluation

**Scaling Study.** For the transition from Base to Large size, we pre-train our model on ImageNet-21K and fine-tune it on ImageNet-1K via supervised learning. When scaling from Large to Huge, we utilize the DataComp1B [10] dataset within a vision-language pre-training paradigm, allowing us to study the effects of scaling the model to a massive size. For evaluation, we evaluate the zero-shot accuracy of these models on ImageNet-1K.

Table 1: Top-1 accuracy of CRATE-$\alpha$ on ImageNet-1K with different model scales when pre-trained on ImageNet-21K and then fine-tuned on ImageNet-1K. For comparison, we also list the results from the paper [46] which demonstrate the diminished return from CRATE base to large, trained only on ImageNet-1K. "IN-21K" refers to ImageNet-21K. ([‡]Results from [46].)

| Models (Base) | ImageNet-1K(%) | Models (Large) | ImageNet-1K(%) |
|---|---|---|---|
| CRATE-B/16 w/o IN-21K | 70.8[‡] | CRATE-L/16 w/o IN-21K | 71.3[‡] |
| CRATE-$\alpha$-B/32 | 76.5 | CRATE-$\alpha$-L/32 | 80.2 |
| CRATE-$\alpha$-B/16 | 81.2 | CRATE-$\alpha$-L/14 | 83.9 |
| CRATE-$\alpha$-B/8 | 83.2 | CRATE-$\alpha$-L/8 | 85.1 |

**Downstream Applications.** We include additional experimental results on four downstream datasets (CIFAR-10/100, Oxford Flowers, and Oxford-IIT Pets). We also examine the dense prediction capability of CRATE-$\alpha$ by training it on segmentation tasks using the ADE20K dataset [51]. For language tasks, we conduct new experiments with CRATE-$\alpha$ using autoregressive training on OpenWebText, following the setup in nanoGPT [16].

**Interpretability**. Following the evaluation setup of CRATE as outlined in [46], we apply MaskCut [43] to validate and evaluate the rich semantic information captured by our model in a zero-shot setting, including both qualitative and quantitative results.

## 4.2   Training and Fine-tuning Procedures

**Scaling Study. (1) Base to Large size:** We initially pre-train the CRATE-$\alpha$ model on ImageNet-21K and subsequently fine-tune it on ImageNet-1K. During the pre-training phase, we set the learning rate to $8 \times 10^{-4}$, weight decay to 0.1, and batch size to 4096. We apply data augmentation techniques such as Inception crop [35] resized to 224 and random horizontal flipping. In the fine-tuning phase, we adjust the base learning rate to $1.6 \times 10^{-4}$, maintain weight decay at 0.1, and batch size at 4096. We apply label smoothing with a smoothing parameter of 0.1 and apply data augmentation methods including Inception crop, random horizontal flipping, and random augmentation with two transformations (magnitude of 9). For evaluation, we resize the smaller side of an image to 256 while maintaining the original aspect ratio and then crop the central portion to 224×224. In both the pre-training and fine-tuning phases, we use the AdamW optimizer [24] and incorporate a warm-up strategy, characterized by a linear increase over 10 epochs. Both the pre-training and fine-tuning are conducted for a total of 91 epochs, utilizing a cosine decay schedule.

**(2) Large to Huge size:** In the pre-training stage, we utilize an image size of 84×84, and the maximum token length is 32, with a total of 2.56 billion training samples. During the fine-tuning stage, we increase the image size to 224×224 while maintaining the maximum token length at 32, with a 512 million training samples. Here, the key distinction between the pre-training stage and the fine-tuning stage is the image size. A smaller image size results in a faster training speed. In the configurations of CRATE-$\alpha$-CLIPA-B, CRATE-$\alpha$-CLIPA-L, and CRATE-$\alpha$-CLIPA-H, we use the CRATE-$\alpha$ model as the vision encoder, and utilize the same pre-trained huge transformer model from CLIPA [18] as the text encoder. For both the pre-training and fine-tuning stages, we freeze the text encoder and only train the vision encoder, i.e., the CRATE-$\alpha$ model. As we will show in the later results, this setup effectively demonstrates the scaling behaviors of CRATE-$\alpha$ models in the image domain. Detailed hyperparameter settings can be found in Appendix A.

**Downstream Applications.** On four downstream datasets, we follow the training setup from [46]. For the segmentation task, we compare the performance of CRATE and CRATE-$\alpha$ on the ADE20K dataset, mainly following the setup of [30] with minor modifications. Our batch size is set to 128, and the total number of training steps is 5000. For the language task, we conduct experiments with CRATE-$\alpha$ using autoregressive training on OpenWebText, following the setup in [16]. We compare CRATE-$\alpha$ models (57M and 120M) with CRATE and GPT-2, using results from CRATE reported in [45].

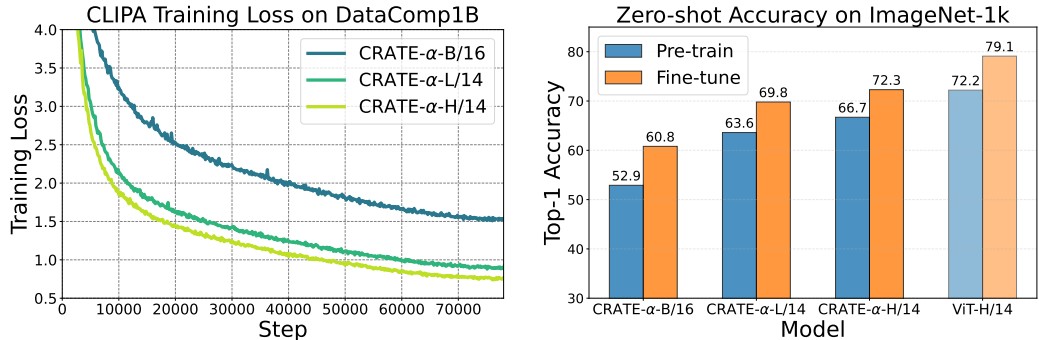

Figure 4: (*Left*) Comparing training loss curves of CRATE-$\alpha$-CLIPA with different model sizes on DataComp1B. (*Right*) Comparing zero-shot accuracy of CRATE-$\alpha$-B/L/H models and ViT-H on ImageNet-1K.

### 4.3 Results and Analysis

**Scaling the CRATE-$\alpha$ Model from Base to Large.** As shown in Table 1, we compare CRATE-$\alpha$-B and CRATE-$\alpha$-L at patch sizes 32, 16, and 8. Firstly, we find our proposed CRATE-$\alpha$-L consistently achieves significant improvements across all patch sizes. Secondly, compared with the results of the vanilla CRATE (the first row of Table 1), increasing from CRATE-B to CRATE-L results in only a 0.5% improvement on ImageNet-1K. This indicates a case of diminishing returns. These findings compellingly highlight that the scalability of CRATE-$\alpha$ models significantly outperforms that of the vanilla CRATE. Meanwhile, the training loss in the pre-training stage is presented in Fig. 3; as the model capacity increases, the trend of the training loss improves predictably. This phenomenon is also described in [9].

**Scaling the CRATE-$\alpha$ Model from Large to Huge.** From the results shown in Fig. 4, we find that: (1) CRATE-$\alpha$-CLIPA-L/14 significantly outperforms CRATE-$\alpha$-CLIPA-B/16 by 11.3% and 9.0% in terms of ImageNet-1K zero-shot accuracy during the pre-training and fine-tuning stages, respectively. The substantial benefit suggests that the quality of learned representation may be constrained by the model size. Therefore, increasing the model size effectively leverages larger amounts of data. (2) When continuing to scale up model size, we also observe that CRATE-$\alpha$-CLIP-H/14 continues to benefit from larger training datasets, outperforming CRATE-$\alpha$-CLIP-L/14 by 3.1% and 2.5% in terms of ImageNet-1K zero-shot accuracy during the pre-training and fine-tuning stages, respectively. This demonstrates the strong scalability of the CRATE-$\alpha$ model. To explore the performance ceiling, we train a standard ViT-CLIPA-H/14 from scratch and observe improved performance.

**Downstream Applications.** On four downstream datasets, as shown in Table 2, we find that CRATE-$\alpha$ consistently outperforms CRATE, with both models pre-trained on IN21K, while CRATE-$\alpha$ demonstrates improved performance as model size increases. For the segmentation task, results in Table 3 show that CRATE-$\alpha$ consistently outperforms CRATE across all key metrics, with both models pre-trained on IN21K. These findings indicate significant performance gains in vision tasks beyond classification. For the language task, Table 4 shows that CRATE-$\alpha$ significantly improves over CRATE in language modeling. Due to limited time and resource constraints, we completed 80% of the total iterations for CRATE-$\alpha$-small and 55% for CRATE-$\alpha$-base, compared to the 600K total iterations used for CRATE. Nevertheless, CRATE-$\alpha$ still demonstrated notable improvements.

**Interpretability.** As shown in Fig. 5, we provide the segmentation visualization on COCO val2017 [20] for CRATE-$\alpha$, CRATE, and ViT, respectively. We find that our model preserves and even improves the (semantic) interpretability advantages of CRATE. Moreover, we summarize quantitative evaluation results on COCO val2017 in Table 6. Interestingly, when scaling up model size for CRATE-$\alpha$, the Large model improves over the Base model in terms of object detection and segmentation.

### 4.4 Compute-efficient Scaling Strategy

We further explore methods to scale models efficiently in terms of computation. Table 1 demonstrates that the CRATE-$\alpha$ model scales effectively from the Base model to its larger variants. However, the pre-training computation for the top-performing model, CRATE-$\alpha$-L/8, is resource-intensive on

Table 2: The performance comparison between CRATE and CRATE-$\alpha$ across various datasets.

| Dataset | CRATE-B/32 | CRATE-$\alpha$-B/32 | CRATE-$\alpha$-L/32 | CRATE-$\alpha$-B/16 | CRATE-$\alpha$-L/14 |
|---|---|---|---|---|---|
| CIFAR-10 | 97.22 | 98.17 | 98.68 | 98.67 | 99.10 |
| CIFAR-100 | 85.27 | 89.40 | 91.16 | 90.58 | 92.57 |
| Oxford Flowers-102 | 93.90 | 97.77 | 99.01 | 99.27 | 99.56 |
| Oxford-IIIT-Pets | 80.38 | 88.19 | 90.46 | 92.70 | 93.98 |

Table 3: Performance comparison of CRATE models with different configurations.

| Model | Scope | mIoU | mAcc | aAcc |
|---|---|---|---|---|
| CRATE-$\alpha$-B/32 | global | 35.35 | 45.28 | 77.63 |
| CRATE-B/32 | global | 30.28 | 39.29 | 75.21 |

Table 4: The comparison between CRATE and CRATE-$\alpha$ on the NLP task using the OpenWebText dataset.

| | GPT-2-base | CRATE-base | CRATE-$\alpha$-small | CRATE-$\alpha$-base |
|---|---|---|---|---|
| Model size | 124M | 60M | 57M | 120M |
| Cross-entropy validation loss | 2.85 | 3.37 | 3.28 | 3.14 |

Table 5: Compute-efficient scaling strategy. To reduce the compute requirements of the pre-training stage, we use a model with a larger patch size. This results in a shorter token length for the same input size. The second and fourth columns indicate the compute requirements for the pre-training and fine-tuning stages, respectively, measured in TPU v3 core-hours. Details are provided in Section 4.4.

| Pre-train | Core-hours | Fine-tune | Core-hours | Total core-hours | IN-1K(%) |
|---|---|---|---|---|---|
| CRATE-$\alpha$-L/32 | 2,652 | CRATE-$\alpha$-L/14 | 872 | 3,524 | 83.7 |
| | | CRATE-$\alpha$-L/8 | 3,486 | 6,138 | 84.2 |
| CRATE-$\alpha$-L/14 | 8,947 | CRATE-$\alpha$-L/14 | 872 | 9,819 | 83.9 |
| CRATE-$\alpha$-L/8 | 35,511 | CRATE-$\alpha$-L/8 | 3,486 | 38,997 | 85.1 |

ImageNet-21K. Inspired by CLIPA [18], we aim to reduce computational demands by using reduced image token sequence lengths, while maintaining the same training setup during the fine-tuning stage. The results are summarized in Table 5.

**Results and analysis.** (1) When fine-tuning with CRATE-$\alpha$-L/14 and using CRATE-$\alpha$-L/32 for pre-training on ImageNet-21K, this approach consumes about 35% of the TPU v3 core-hours required by CRATE-$\alpha$-L/14, yet achieves a promising 83.7% top-1 accuracy on ImageNet-1K, comparable to the 83.9% achieved by CRATE-$\alpha$-L/14; (2) When fine-tuning with CRATE-$\alpha$-L/8 and using CRATE-$\alpha$-L/32 for pre-training, this approach consumes just 15% of the training time required by CRATE-$\alpha$-L/8, yet it still achieves a promising 84.2% top-1 accuracy on ImageNet-1K, compared to 85.1% when using the CRATE-$\alpha$-L/8 model in the pre-training stage; (3) While the total computational cost of CRATE-$\alpha$-L/32 + CRATE-$\alpha$-L/8 is less than that of CRATE-$\alpha$-L/14 + CRATE-$\alpha$-L/14, the performance of the former is slightly better. In summary, we find that this strategy offers a valuable reference for efficiently scaling CRATE-$\alpha$ models in the future.

## 5 Discussion

**Limitations.** Although we have used some existing compute-efficient training methods (e.g., CLIPA [18]) and have initiated an exploration into compute-efficient scaling strategies for white-box transformers in Section 4.4, this work still requires a relatively large amount of computational resources, which may not be easily accessible to many researchers.

**Societal impact.** A possible broader implication of this research is the energy consumption needed to conduct the experiments in our scaling study. However, there is growing interest in developing white-box transformers for better interpretability and transparency across a wide range of tasks and domains, including image segmentation [46], self-supervised masked autoencoders [26], and integrated sensing and communications [50], etc. Moreover, our results on the scalability of white-

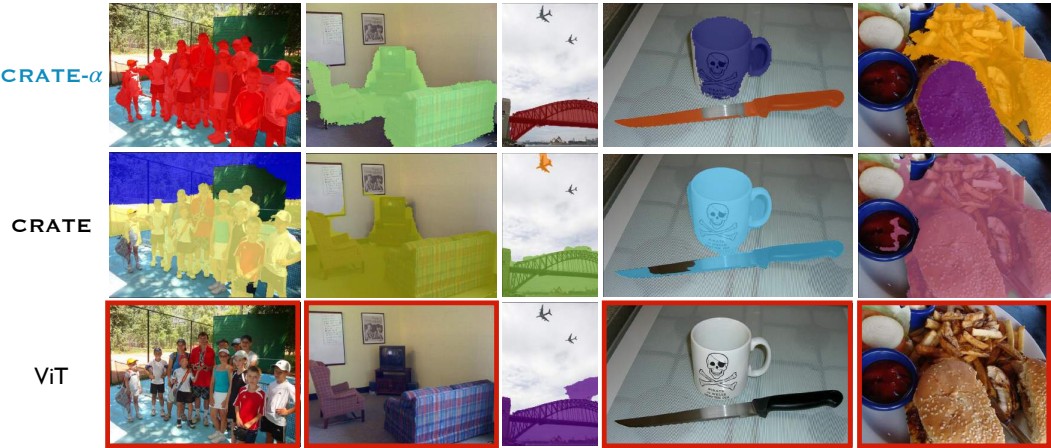

Figure 5: **Visualization of segmentation on COCO val2017 [20] with MaskCut [43].** (*Top row*) Supervised CRATE-$\alpha$ effectively identifies the main objects in the image. Compared with CRATE (*Middle row*), CRATE-$\alpha$ achieves better segmentation performance in terms of boundary. (*Bottom row*) Supervised ViT fails to identify the main objects in most images. We mark failed image with □.

Table 6: **Object detection and fine-grained segmentation via MaskCut on COCO val2017 [20].** We evaluate models of various scales and assess their average precision using COCO's official evaluation metric. Compared with existing models such as CRATE and ViT, CRATE-$\alpha$ model achieves a notable performance gain. In addition, when scaling CRATE-$\alpha$ from base to large, it also exhibits the benefit of scalability.

| Model | Train | Detection | | | Segmentation | | |
|---|---|---|---|---|---|---|---|
| | | $AP_{50} \uparrow$ | $AP_{75} \uparrow$ | $AP \uparrow$ | $AP_{50} \uparrow$ | $AP_{75} \uparrow$ | $AP \uparrow$ |
| CRATE-B/8 [47] | Supervised | 2.9 | 1.0 | 1.3 | 2.2 | 0.7 | 1.0 |
| ViT-B/8 [47] | Supervised | 0.8 | 0.2 | 0.4 | 0.7 | 0.5 | 0.4 |
| CRATE-$\alpha$-B/8 | Supervised | 3.5 | 1.1 | 1.5 | 2.2 | 1.0 | 1.1 |
| CRATE-$\alpha$-L/8 | Supervised | 4.0 | 1.7 | 2.0 | 2.7 | 1.1 | 1.4 |

box transformers could also shed light on scaling up a broader class of white-box deep neural networks, such as white-box ISTA networks and their variants [11, 34, 3, 48, 17], designed via unrolled optimization. In summary, we believe that our findings and insights could be helpful for developing white-box transformers for a wide range of applications and tasks, benefiting a broad audience interested in building more interpretable and performant deep learning models and further amortizing the pre-training compute costs.

## 6 Conclusion

This paper provides the first exploration of training white-box transformer CRATE at scale for vision tasks. We introduce both principled architectural changes and improved training recipes to unleash the potential scalability of the CRATE type architectures. With these modifications, we successfully scale up the CRATE-$\alpha$ model along both the dimensions of model size and data size, while preserving, in most cases even improving, the semantic interpretability of the learned white-box transformer models. We believe this work provides valuable insights into scaling up mathematically interpretable deep neural networks, not limited to transformer-like architectures.

**Acknowledgement**

This work is supported by a gift from Open Philanthropy, TPU Research Cloud (TRC) program, and Google Cloud Research Credits program.

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

# Appendix

## A   Additional Experiments and Details

### A.1   Model configuration.

We provide details about CRATE-$\alpha$ model configurations in Table 7.

Table 7: Model configurations for different sizes of CRATE-$\alpha$, parameter counts, and comparisons to CRATE models. $L$ is depth, $d$ is the hidden size, and $K$ is the number of heads.

| Model Size | $L$ | $d$ | $K$ | CRATE-$\alpha$ # Params | CRATE # Params |
|---|---|---|---|---|---|
| Tiny | 12 | 192 | 3 | 4.8M | 1.7M |
| Small | 12 | 576 | 12 | 41.0M | 13.1M |
| Base | 12 | 768 | 12 | 72.3M | 22.8M |
| Large | 24 | 1024 | 16 | 253.8M | 77.6M |
| Huge | 32 | 1280 | 16 | 526.8M | 159.8M |

Table 8: The comparison between CRATE-$\alpha$ and ViT. FLOPs and throughput are calculated based on an input size of 224x224 on an NVIDIA RTX A6000 graphics card.

| Model | FLOPs (G) | #Params (M) | Throughput | Model | FLOPs (G) | #Params (M) | Throughput |
|---|---|---|---|---|---|---|---|
| CRATE-$\alpha$-B/32 | 6.4 | 74.0 | 499 | ViT-B/32 | 4.4 | 88.2 | 706 |
| CRATE-$\alpha$-B/16 | 25.8 | 72.3 | 233 | ViT-B/16 | 17.6 | 86.5 | 375 |
| CRATE-$\alpha$-L/32 | 22.8 | 256.0 | 215 | ViT-L/32 | 15.4 | 306.5 | 329 |
| CRATE-$\alpha$-L/14 | 119.7 | 253.7 | 56 | ViT-L/14 | 81.1 | 304.1 | 85 |

### A.2   Comparison of model structure with ViT.

We also compare CRATE-$\alpha$ to ViT in terms of computational costs, number of parameters, and inference speed. These comparisons are summarized in Table 8, where CRATE-$\alpha$ matches ViT's efficiency while achieving similar accuracy. With the same number of layers and embedding dimensions, CRATE-$\alpha$ has fewer parameters than ViT, and its FLOPs/Throughput is slightly higher.

To more accurately compare CRATE-$\alpha$ and ViT with larger model sizes, we conduct experiments on CRATE-$\alpha$-L/16 with an image resolution of 336, nearly matching the setup of ViT-L/16. Both models use a similar amount of FLOPs: 210G for CRATE-$\alpha$-L/16 compared to 191G for ViT-L/16. The throughput, or images processed per second, is also comparable at 35.53 for our model versus 35.56 for ViT-L/16. The accuracy of CRATE-$\alpha$-L/16 reach 84.6%, closely approaching ViT's 85.2% under similar conditions. Meanwhile, combining the trend from Figure 1 (right) in the main paper, this narrowing performance gap from Base to Large model size suggests that CRATE-$\alpha$ can nearly match ViT's performance in large-scale settings. Besides, CRATE-$\alpha$ inherits the mathematical interpretability of the white-box models and can also achieve much better semantic interpretability evaluated by zero-shot segmentation.

### A.3   Training details of CRATE-$\alpha$-CLIPA models.

When employing the CRATE-$\alpha$ architecture to replace the vision encoder in the CLIPA [18] framework, we essentially follow the original CLIPA training recipe. The setup for the pre-training stage is presented in Table 9. During the fine-tuning stage, we made some modifications: the input image size is set to $224 \times 224$, the warmup steps are set to 800, and the base learning rate is set to 4e-7. When calculating the loss, we use the classification token from the vision encoder as the image feature and the last token from the text encoder as the text feature.

To explore the performance ceiling, we also train a ViT-CLIPA model from scratch. Most of the hyperparameters remain the same as those in Table 9, but there are some modifications in the pre-training stage. The batch size is set to 65,536, and the text length is set to 8 to speed up training. As

with the CLIPA setup, warm-up steps are set to 3,200. Additionally, we add color jitter and grayscale augmentation, and use global average pooling instead of the classification token. These modifications help stabilize training.

| Config | Value |
|---|---|
| optimizer | AdamW [25] |
| optimizer momentum | (0.9, 0.95) |
| batch size | 32768 |
| base lr | 8e-6 |
| minimal lr | 0 |
| warm-up steps | 1600 |
| schedule | cosine decay [23] |
| weight decay | 0.2 |
| random crop area | (40, 100) |
| resize method | bi-linear |
| temperature init | 1/0.07 [13, 19] |

Table 9: **Pre-training hyper-parameters for CLIPA.**

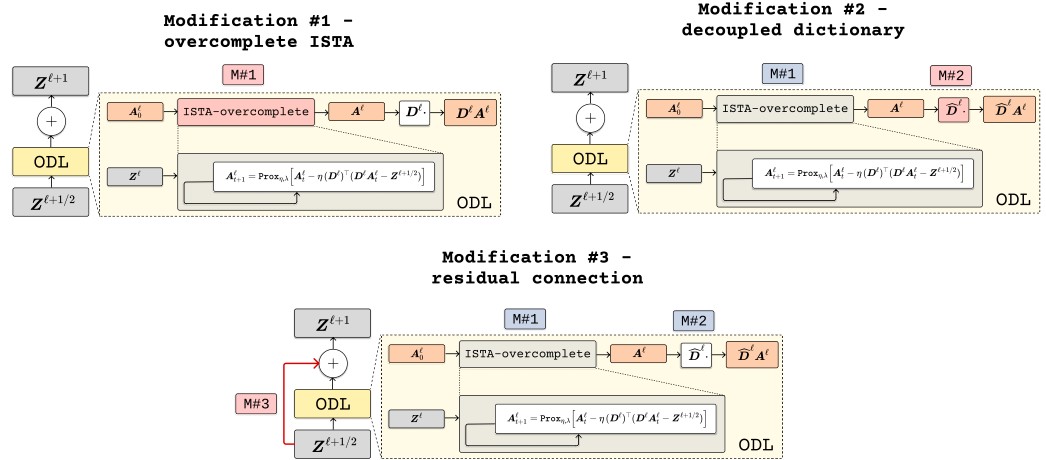

Figure 6: One layer of the CRATE-$\alpha$ model architecture (with more details for the three modifications described in Section 3.

**Visualization of self-attention maps of CRATE-$\alpha$.** We provide visualization of attention maps of CRATE-$\alpha$ in Fig. 7.

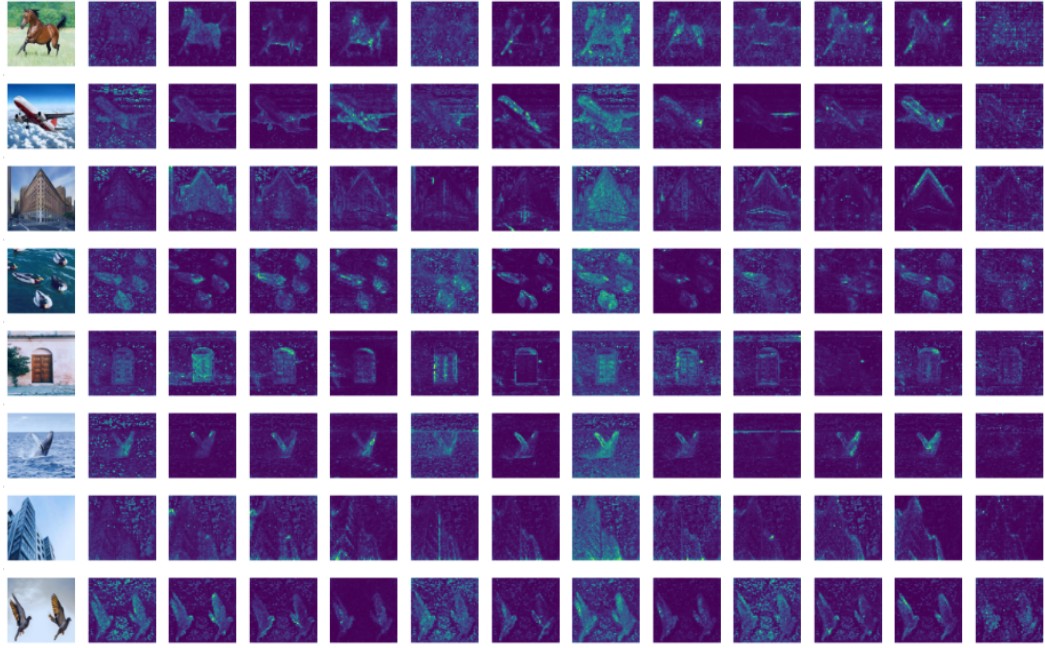

Figure 7: We visualize the self-attention maps of the CRATE-$\alpha$ Base model using $8 \times 8$ patches trained using classification. Similar to the original CRATE [47], our model also demonstrates the capability to automatically capture the structural information of objects. For each row, the original image is displayed on the left, while the corresponding self-attention maps are shown on the right. The number of self-attention maps corresponds to the number of heads in the CRATE-$\alpha$ model.

**Visualization of loss curves.** We visualize the training loss curves of the four models, including CRATE and its three variants, in Fig. 8. We visualize the training loss curves of CRATE-$\alpha$-Base with different patch sizes in Fig. 9. In Fig. 10, we also visualize the training loss curves of models trained with efficient scaling strategy described in Section 4.4 in the main paper.

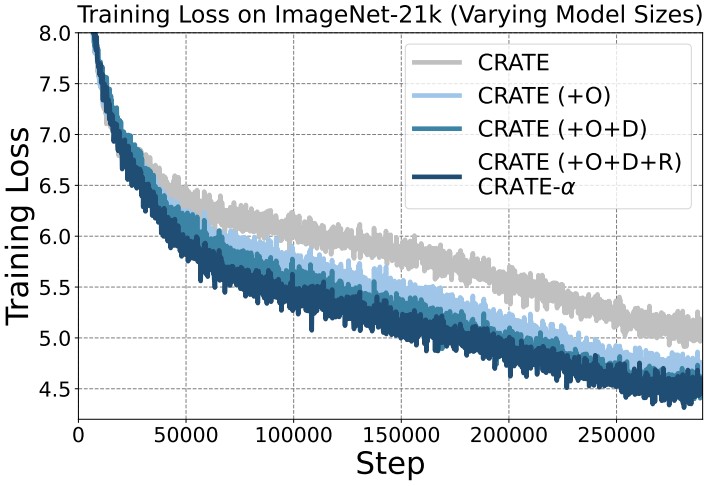

Figure 8: Training loss curves of different model architectures (mentioned in Fig. 1 in the main paper) on ImageNet-21K. The patch size is 32 for all four models shown in this figure. (+O: +overcomplete dictionary, +D: +decoupled dictionary, +R: +residual connection.)

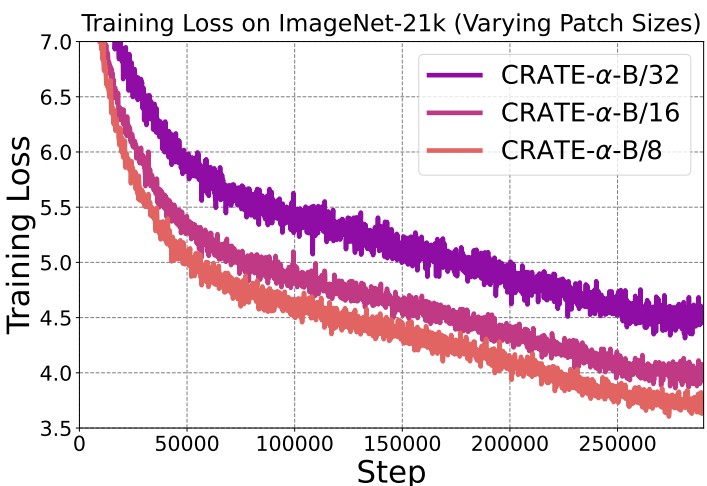

Figure 9: Comparing training loss curves across CRATE-$\alpha$-Base with different patch sizes.

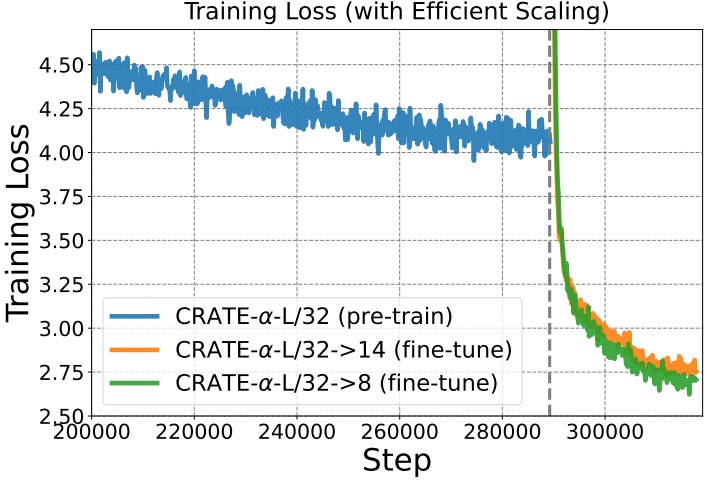

Figure 10: Comparing training loss curves when using the efficient scaling strategy. The blue curve corresponds to the CRATE-$\alpha$-Large/32 model (in the pre-training stage). After pre-training the CRATE-$\alpha$-Lage/32, we further fine-tune it with smaller patch sizes (longer token length), including patch size 14 (orange curve) and patch 8 (green curve).

