# OpenReview forum: "Scaling White-Box Transformers for Vision"
_NeurIPS.cc/2024/Conference — NeurIPS 2024 poster_

### Official Review · Reviewer_YH3U · 2024-07-11

**Soundness:** 3
**Presentation:** 3
**Contribution:** 3
**Rating:** 7
**Confidence:** 4

**Summary:**

This paper introduces CRATE-α, an enhanced variant of the CRATE (Coding RATE Transformer) architecture, designed to scale efficiently while maintaining mathematical interpretability. The authors address the open question of CRATE's scalability by proposing strategic modifications to the sparse coding block and a refined training recipe. Extensive experiments demonstrate CRATE-α's effectiveness, showcasing improved performance on ImageNet classification tasks compared to the original CRATE model. Notably, the CRATE-α-B model achieved an 83.2% accuracy rate, a significant improvement over the previous best CRATE-B model.

**Strengths:**

The paper presents a novel architecture, CRATE-α, that builds upon the existing CRATE model with minimal yet strategic modifications, enhancing scalability without compromising interpretability.

The authors provide a wealth of empirical evidence supporting the effectiveness of CRATE-α, including comparative results on ImageNet classification tasks and a detailed analysis of training behaviors across different model scales.

A key strength is the paper's focus on maintaining the interpretability of the model, which is often a trade-off in scaling deep learning models. The authors demonstrate that CRATE-α models retain high-quality unsupervised object segmentation capabilities.

The paper includes a thorough exploration of scaling behaviors, from Base to Large to Huge model sizes, using both supervised learning on ImageNet and vision-language pre-training with contrastive learning on DataComp1B.

**Weaknesses:**

Could the proposed architecture work well on other tasks like NLP?


While the paper provides a detailed analysis of the model's performance on ImageNet, there might be a need for more discussion on how these results generalize to other datasets and real-world applications.

**Questions:**

See the weakness.

**Limitations:**

Yes

---

> ### Author Rebuttal · Authors · 2024-08-07
>
> Thank you for your review. Below we attempt to resolve the questions you posed.
>
> >**Q1**: *Could the proposed architecture work well on other tasks like NLP?*
>
> **A1**: Thank you for your suggestions on new experiments on NLP. Please refer to our response to '**Q3: Performance of CRATE-α on NLP task**' in our common response.
>
> >**Q2**: *While the paper provides a detailed analysis of the model's performance on ImageNet, there might be a need for more discussion on how these results generalize to other datasets and real-world applications.*
>
> **A2**: Thank you for your suggestions on new experiments on other datasets and real-word applications. Please refer to our response to '**Q2: Additional experimental results on real-world downstream tasks**' in our common response.
>
> We again thank you for your review, and hope we have provided satisfactory responses to your questions. Please let us know if you have further questions or comments.

---

### Official Review · Reviewer_cPqJ · 2024-07-12

**Soundness:** 3
**Presentation:** 2
**Contribution:** 2
**Rating:** 6
**Confidence:** 4

**Summary:**

This paper explores how to train white-box Transformers at scale for visual tasks. The authors propose a new model architecture called CRATE-$\alpha$, which extends the sparse coding block of the original CRATE model. A series of CRATE-$\alpha$ models were trained with varying model sizes, data sizes, and patch sizes using optimized training recipes. The main experiments focus on supervised classification and contrastive CLIP learning, with additional demonstrations of unsupervised semantic segmentation capability.

**Strengths:**

**Originality:** The paper continues the white-box design philosophy of the original CRATE model while integrating advanced techniques such as overparameterized sparse coding, decoupled dictionary, and residual connections. Although some of these techniques have been previously validated, successfully combining them with a white-box Transformer is a noteworthy achievement. The integration not only works effectively but also yields commendable results.

**Quality:** The paper is technically sound overall, employing rigorous notation and formula definitions to elucidate the design principles. The proposed models demonstrate significant improvements compared to the previous generation of CRATE models. Additionally, the authors are careful and honest in evaluating the weaknesses and limitations of their work.

**Weaknesses:**

**Clarity:**
- The paper is heavily symbolized, relying extensively on intricate mathematical formulations rather than clear diagrams and straightforward language. Although this maintains academic rigor and professionalism, it severely hampers understanding of the paper's details and the broader dissemination of the model. Incorporating corresponding illustrations to explain the three modifications and comparing them with the standard Transformer structure would be beneficial.
- The organization of Section 4 is not concise, making it easy for readers to lose track.
  - The distinction between the paragraphs "Dataset and Evaluation" and "Training & Fine-tuning" is not well-defined, especially with the scattered descriptions of the data used.
  - The frequent interleaving of experimental setup descriptions with the presentation of experimental results disrupts the flow and coherence of the narrative.

**Significance:**
- Although CRATE-$\alpha$ shows significant improvements over the original CRATE model, it still lags behind the state-of-the-art. For example, in the right side of Figure 1, CRATE-$\alpha$ typically requires nearly double the training FLOPs to achieve the same accuracy as ViT.
- If the scalability and interpretability of a white-box Transformer architecture does not offer substantial insights and improvements, practitioners might prefer models with stronger performance but lower interpretability.

**Questions:**

1. As previously mentioned, as shown on the right side of Figure 1, CRATE-$\alpha$ usually requires approximately twice the FLOPs to reach the performance level of ViT, putting it at a noticeable disadvantage.

2. How does the performance improvement of CRATE-$\alpha$ compare to the original CRATE? Neither the CRATE models in Table 1 nor Figure 1 were pretrained on ImageNet-21K. Why was this not included for a fair comparison?

3. Lines 232-233 and Figure 3 describe the model’s **training loss** as predictable. Why not the **validation loss**, which is the primary concern in scaling laws and practical applications?

4. Table 2 only shows the compute requirements for the **pre-training stage**. Why does it not include the **fine-tuning** stage? Considering the total computational effort, I would like to see a comparison of the two scaling strategies: *CRATE-$\alpha$-L/32 + CRATE-$\alpha$-L/8* versus *CRATE-$\alpha$-L/14 + CRATE-$\alpha$-L/14*.

5. How was the amount of training data determined? Was there a specific standard or a FLOPs constraint? For example:
   - In Section 4.1, for training models from Base to Large, both pre-training and fine-tuning were conducted for a total of **91** epochs.
   - In Section 4.1, for training models from Large to Huge, there were **2.56** billion and **512** million training samples, respectively.

**Limitations:**

The authors adequately addressed the limitations.

---

> ### Author Rebuttal · Authors · 2024-08-07
>
> Thank you for your review. Below we attempt to resolve the questions you posed.
>
> >**Q1**: *The paper is heavily symbolized, ... it severely hampers understanding of the paper's details ...*
>
> **A1**: Thank you for the paper presentation suggestion. We've added a new diagram to our rebuttal pdf (Figure 1 in the rebuttal pdf) illustrating the three modifications. In the revised version, we'll include additional figures/diagrams to clarify these modifications and enhance explanations in Section 3, improving our presentation.
>
> >**Q2**: *The organization of Section 4 is not concise, making it easy for readers to lose track.*
>
> **A2**: Thank you for your suggestion on the paper presentation. In our revision, we will reorganize Section 4 to further improve the clarity, including providing an overview of the datasets used in the paper in a paragraph, and adding pointers to different figures/tables when introducing the datasets.
>
> >**Q3**: *Although CRATE-α shows significant improvements over the original CRATE model, it still lags behind the state-of-the-art.*
>
> **A3**: Please see our response to '**Q1: Comparison with ViT**' in the common response, indicating that under similar FLOPs, large CRATE-α-L/16 models nearly match ViT-L/16 performance.
>
> >**Q4**: *If the scalability and interpretability of a white-box Transformer architecture does not offer substantial insights and improvements, practitioners might prefer models with stronger performance but lower interpretability.*
>
> **A4**: We agree that performance/accuracy is an important metric for evaluating different models. However, we believe that interpretability is also an important property in machine learning models built for real-world applications, where reliability and trustworthiness are paramount, such as self-driving and robotics (among a long list of others). As we have shown in this work (including new experimental results described in **A3**), it is very possible to scale up white-box models that are designed from principles to achieve the same performance as black-box ones. Since, as we have shown, we can train white-box models with significantly improved mathematical and semantic interpretability properties compared to the usual transformer at negligible performance drop, we believe that our model may be well-suited for real-world applications in which such properties are necessary.
> We will add a paragraph on discussing the trade-off between interpretability and performance to Section 5 Discussion in our revised version.
>
> > **Q5**: *As previously mentioned, as shown on the right side of Figure 1, CRATE-α usually requires approximately twice the FLOPs to reach the performance level of ViT, putting it at a noticeable disadvantage.*
>
> **A5**: see our response in **A3**.
>
> >**Q6**: *How does the performance improvement of CRATE-α compare to the original CRATE? Neither the CRATE models in Table 1 nor Figure 1 were pretrained on ImageNet-21K. Why was this not included for a fair comparison?*
>
> **A6**: In Figure 1 (left), **all four model variants are pretrained on IN21K and then fine-tuned on IN1K**. We can see that CRATE-α-B/32 (76.5% on IN1K) outperforms CRATE-B/32 (68.5% on IN1K) by 8.5%. We also tried training the original CRATE-L model on 21K, but the training was unstable and we found it is challenging to scale CRATE-L to IN21K. The CRATE-L model cannot be effectively scaled to larger datasets due to unstable optimization. This is also the motivation for proposing the CRATE-α model.
>
> >**Q7**: *Lines 232-233 and Figure 3 describe the model’s training loss as predictable. Why not the validation loss, which is the primary concern in scaling laws and practical applications?*
>
> **A7**: Thank you for your suggestion. We have plotted the validation loss for Figure 3, the validation loss curves follow the similar trend as the training loss curves in Figure 3. We will add these new results to our revised version.
>
> >**Q8**: *Table 2 only shows the compute requirements for the pre-training stage. Why does it not include the fine-tuning stage? Considering the total computational effort, I would like to see a comparison of the two scaling strategies: CRATE-α-L/32 + CRATE-α-L/8 versus CRATE-α-L/14 + CRATE-α-L/14.*
>
> **A8**: Thank you for your suggestions regarding the comparison. Given the higher number (10x) of training samples in pre-training compared to fine-tuning, pre-training dominates compute costs. Following your advice, we've summarized the total cost of pre-training and fine-tuning in Table 4 of our rebuttal pdf file. While the total computational cost of CRATE-α-L/32 + CRATE-α-L/8 is less than that of CRATE-α-L/14 + CRATE-α-L/14, the performance of the former is slightly better. This reinforces the benefit of reducing image token sequence lengths during pre-training for compute-efficient scaling.
>
> > **Q9**: *How was the amount of training data determined? Was there a specific standard or a FLOPs constraint?*
>
> **A9**: Regarding the number of training epochs, we mainly follow the setup in previous work: [YCT+2024], [YBP+2023], and [TCJ2022]. Specifically, C.1 section in paper [YCT+2024], Table 13 of C.5 in paper [YBP+2023], and section 4.1 in paper [TCJ2022]. For the number of training samples from large to huge, we referred to Table 1 of [LWX2023].
>
> We again thank you for your review, and hope we have provided satisfactory responses to your questions. Please let us know if you have further questions or comments.
>
> [YCT+2024] Emergence of Segmentation with Minimalistic White-Box Transformers. Yaodong Yu, Tianzhe Chu, Shengbang Tong, et al. CPAL 2024.
>
> [YBP+2023] White-Box Transformers via Sparse Rate Reduction: Compression Is All There Is? Yaodong Yu, Sam Buchanan, Druv Pai, et al. arXiv:2311.13110. 2023.
>
> [TCJ2022] DeiT III: Revenge of the ViT. Hugo Touvron, Matthieu Cord, Hervé Jégou. ECCV 2022.
>
> [LWX2023] An Inverse Scaling Law for CLIP Training. Xianhang Li, Zeyu Wang, Cihang Xie. NeurIPS 2023.

---

> > ### Comment · Reviewer_cPqJ · 2024-08-12
> >
> > Regarding A3 and A5, the authors claim that *"that under similar FLOPs, large CRATE-α-L/16 models nearly match ViT-L/16 performance."*  However, this assertion is not convincing when considered alongside Figure 1 (Right) and the response to 'Q1: Comparison with ViT' in the common response.
> >
> > - At approximately 200G FLOPs, CRATE-α-L/16 achieves 84.6%, which means a further 7% increase is needed to match ViT's 85.2%. This gap should not be overlooked. The authors suggest that *"this narrowing performance gap from Base to Large model size suggests that CRATE-α can nearly match ViT’s performance in large-scale settings"*; however, it is observed in Figure 1 (Right) that as the model size increases, achieving such gains becomes increasingly challenging. For instance, CRATE-α-L/8 requires nearly **double** the FLOPs to achieve performance comparable to ViT-L/16.
> >
> > - At smaller scales, CRATE-α appears to keep pace with ViT more easily, reaching comparable performance with a lower **absolute increase** in FLOPs. However, proportionally, CRATE-α consistently requires nearly **double** the increase in FLOPs to match ViT’s performance, as seen in the comparisons between CRATE-α-B/16 and ViT-B/32, as well as CRATE-α-L/14 and ViT-B/16.
> >
> > Regarding A4, the authors reiterate that *"it is very possible to scale up white-box models that are designed from principles to achieve the same performance as black-box ones."* However, I would like to ask what practical **benefits** this white-box design (the so-called interpretability) has, aside from scaling up. For example, if this classification model performs poorly on a certain category, can this interpretability guide me in adjusting the model structure to improve performance in those categories? If the only advantage is scaling up, then such interpretability holds no real benefit, as black-box transformers can scale up more effectively (Figure 1 Right).
> >
> > Regarding A6, the authors' explanation **conflicts** with the paper. Please refer to the last two lines of the Figure 1 caption: *"CRATE is trained only on ImageNet-1K, while CRATE-α and ViT are pre-trained on ImageNet-21K."* Why do the authors now explain that *"all four model variants are pretrained on IN21K and then fine-tuned on IN1K"*? I kindly request the authors to consider my question closely: since CRATE has not been pre-trained on ImageNet-21K while CRATE-α has been *"pre-trained
> > on ImageNet-21K and then fine-tuned on ImageNet-1K"* (as indicated in the caption of Table 1), how can these two models be compared fairly?

---

> ### Author Response · Authors · 2024-08-13
> **Discussion with Reviewer cPqJ (Part 1)**
>
> We are grateful for you engaging with our rebuttal further, and for your critical perspective on the work, which will no doubt improve it. Below we attempt to resolve the questions you posed.
>
> > **Q10**: *At approximately 200G FLOPs, CRATE-α-L/16 achieves 84.6%, which means a further 7% increase is needed to match ViT's 85.2%. This gap should not be overlooked. The authors suggest that "this narrowing performance gap from Base to Large model size suggests that CRATE-α can nearly match ViT’s performance in large-scale settings"; however, it is observed in Figure 1 (Right) that as the model size increases, achieving such gains becomes increasingly challenging. For instance, CRATE-α-L/8 requires nearly double the FLOPs to achieve performance comparable to ViT-L/16. At smaller scales, CRATE-α appears to keep pace with ViT more easily, reaching comparable performance with a lower absolute increase in FLOPs. However, proportionally, CRATE-α consistently requires nearly double the increase in FLOPs to match ViT’s performance, as seen in the comparisons between CRATE-α-B/16 and ViT-B/32, as well as CRATE-α-L/14 and ViT-B/16.*
>
> **A10**: We would like to clarify that **the performance gap between CRATE-α-L/16 (84.6% on IN1K) and ViT-L/16 (85.2% on IN1K) is 0.6%, instead of 7%**. We do agree that there is still a gap between CRATE-α-L/16 and ViT-L/16 (under similar FLOPs), but we would also like to highlight that this gap is much smaller than the previous gap between vanilla CRATE and ViT.
>
> Meanwhile, to the best of our knowledge, there are no public results for ViT-L/8 on a similar setup. It is not clear whether increasing model size from ViT-L/16 to ViT-L/8 could significantly improve the performance of ViT. In particular, the FLOPs of the transformer architecture are closely related to the computation in the self-attention module, which grows quadratically with token length. When the patch size is halved, the token length increases to 4 times compared to the original one, resulting in the computational cost increasing by a factor of 16 for the self-attention module. We believe there is a trade-off between FLOPs consumption and accuracy gains in this context, and the transition from a patch size of 16 to 8 may not be optimal for CRATE-α and ViT.
>
> If we first exclude the model with a patch size of 8 (due to no public available ViT-L/8), we can see that the gap on IN1K **narrows from 4.8%** between *CRATE-α-B/32 (76.5%) and ViT-B/32 (81.3%)* **to 0.6%** between *CRATE-α-L/16 (84.6%) and ViT-L/16 (85.2%)*. The latter group also has similar FLOPs and throughput. This is why we claimed that "this narrowing performance gap from Base to Large model size suggests that CRATE-α can nearly match ViT’s performance in large-scale settings."
>
> We would like to highlight that the main focus of this paper is to compare the differences between CRATE-α and CRATE, and investigate whether it is possible to scale up white-box models and achieve competitive performance. We acknowledge that we did not conduct a comprehensive and thorough comparison between CRATE-α and ViT across various dimensions. Figure 1 (right) did not initially unify these variables, including patch size. Per your suggestion, we will add more experimental results (e.g., including different patch sizes for both ViTs and CRATE-α) in our revised version. While we think this is an interesting direction to explore in the future, it is beyond the main focus of this paper.

---

> ### Author Response · Authors · 2024-08-13
> **Discussion with Reviewer cPqJ (Part 2)**
>
> >**Q11**: *Regarding A4, the authors reiterate that "it is very possible to scale up white-box models that are designed from principles to achieve the same performance as black-box ones." However, I would like to ask what practical benefits this white-box design (the so-called interpretability) has, aside from scaling up. For example, if this classification model performs poorly on a certain category, can this interpretability guide me in adjusting the model structure to improve performance in those categories? If the only advantage is scaling up, then such interpretability holds no real benefit, as black-box transformers can scale up more effectively (Figure 1 Right).*
>
> **A11**: Thank you for your question regarding the practical benefits of this interpretable white-box design. Firstly, we would like to emphasize that model performance (such as accuracy) is not the only metric to evaluate a model. As we described in this work and shown in the prior work on white-box models [GL2010, CLW+2018, ZTA+2019, YBP+2023], we believe building understandings of how deep learning models work is also an important property for building trustworthy models. For example, 'advancing ongoing research in AI safety, including on the interpretability of AI systems' decision-making processes and on increasing the robustness of AI systems against misuse.' from [Commitments2023], with similar messages from as other sources [RSG2016, LSC+2023].
>
> Meanwhile, regarding your question '*For example, if this classification model performs poorly on a certain category, can this interpretability guide me in adjusting the model structure to improve performance in those categories?*', there is a growing literature --- including our work --- on demonstrating interpretability leads to practical benefits across different settings. For example, [YCT+2024] and our work demonstrated that better internal interpretability leads to improved zero-shot segmentation, [MRM+2024] showed that one can leverage interpretability (based on sparse auto-encoders) to improve the accuracy of classifiers in language models, and [GES2024] demonstrated that one can use interpretability as a tool to remove spurious features.
>
> Thank you again for your question on the motivation of building white-box models. Given the white-box/interpretable models are useful for building trustworthy and safe AI systems, our work aims to study whether it is possible to scale up such interpretable models and close their performance gap compared with black-box ones. Per your suggestion, we will add the above discussions on the practical benefits of interpretability to our revised version.
>
>
> [RSG2016] "Why Should I Trust You?": Explaining the Predictions of Any Classifier. Marco Tulio Ribeiro, Sameer Singh, Carlos Guestrin. KDD 2016.
>
> [LSC+2023] On the importance of interpretable machine learning predictions to inform clinical decision making in oncology. Sheng-Chieh Lu, Christine L. Swisher, Caroline Chung, et al. Frontiers in Oncology, 2023.
>
> [Commitments2023] Voluntary AI Commitments. https://www.whitehouse.gov/wp-content/uploads/2023/07/Ensuring-Safe-Secure-and-Trustworthy-AI.pdf
>
> [GL2010] Karol Gregor and Yann LeCun. Learning fast approximations of sparse coding. ICML 2010.
>
> [CLW+2018] Theoretical Linear Convergence of Unfolded ISTA and its Practical Weights and Thresholds. Xiaohan Chen, Jialin Liu, Zhangyang Wang, Wotao Yin. NeurIP S2018.
>
> [ZTA+2019] Deep Network Classification by Scattering and Homotopy Dictionary Learning. John Zarka, Louis Thiry, Tomás Angles, Stéphane Mallat. arXiv:1910.03561. 2019.
>
> [YBP+2023] White-Box Transformers via Sparse Rate Reduction. Yaodong Yu, Sam Buchanan, Druv Pai, Tianzhe Chu, Ziyang Wu, Shengbang Tong, Benjamin David Haeffele, Yi Ma. NeurIPS 2023.
>
> [YCT+2024] Emergence of Segmentation with Minimalistic White-Box Transformers. Yaodong Yu, Tianzhe Chu, Shengbang Tong, Ziyang Wu, Druv Pai, Sam Buchanan, Yi Ma. CPAL 2024.
>
> [GES2024] Interpreting CLIP's image representation via text-based decomposition. Yossi Gandelsman, Alexei A. Efros, Jacob Steinhardt. International Conference on Learning Representations (ICLR), 2024.
>
> [MRM+2024] Sparse Feature Circuits: Discovering and Editing Interpretable Causal Graphs in Language Models. Samuel Marks, Can Rager, Eric J. Michaud, et al. https://arxiv.org/abs/2403.19647. 2024.

---

> ### Author Response · Authors · 2024-08-13
> **Discussion with Reviewer cPqJ (Part 3)**
>
> > **Q12**: Regarding A6, the authors' explanation **conflicts** with the paper. Please refer to the last two lines of the Figure 1 caption: *"CRATE is trained only on ImageNet-1K, while CRATE-α and ViT are pre-trained on ImageNet-21K."* Why do the authors now explain that *"all four model variants are pretrained on IN21K and then fine-tuned on IN1K"*? I kindly request the authors to consider my question closely: since CRATE has not been pre-trained on ImageNet-21K while CRATE-α has been *"pre-trained on ImageNet-21K and then fine-tuned on ImageNet-1K"* (as indicated in the caption of Table 1), how can these two models be compared fairly?
>
> **A12:**  Thank you for your question, we apologize for the confusion by the unclear description in our submission. We would like to clarify that the comparison between CRATE and CRATE-α is fair. Specifically, the caption "CRATE is trained only on ImageNet-1K, while CRATE-α and ViT are pre-trained on ImageNet-21K" **applies only to Figure 1 (right), instead of both left and right figures**.
>
> In particular, we would like to clarify that the training setups for the CRATE model in Figure 1 (left) and Figure 1 (right) are different:
>
> - The results for CRATE-B/16 and CRATE-L/16 in Figure 1 (right) and Table 1 are directly obtained from the original CRATE paper [YBP+2023], where the models were trained only on IN1K. This serves as an intuitive comparison of different published results.
> - The motivation for Figure 1 (left) is to ablate the effectiveness of our proposed three modifications. **It is an ablation study.** Therefore, we kept all training setups the same except for the model architecture. This is the reason — in **A6** of the rebuttal response — we mentioned that "all four model variants are pretrained on IN21K and then fine-tuned on IN1K."
>
> Additionally, according to the results of **Fine-tuning classification datasets** in **Q2: Additional experimental results on real-world downstream tasks** in the common response (we also applied the same training setup here), these results further validate the improvement of CRATE-α compared to CRATE under the same pre-trained dataset (IN21K). In the revision, we will clarify the different training/dataset setups and make sure to deliver clear messages to the readers.
>
> We again thank you for your review and valuable feedback, and hope we have provided satisfactory responses to your questions. Please let us know if you have further questions or comments.
>
>
> [YBP+2023] White-Box Transformers via Sparse Rate Reduction. Yaodong Yu, Sam Buchanan, Druv Pai, et al. NeurIPS 2023.

---

> > ### Comment · Reviewer_cPqJ · 2024-08-13
> >
> > Thank you for the author's prompt clarification; that makes things a bit clearer. However, my initial question and previous comments were specifically concerning Figure 1 (Right) and Table 1, and did not involve Figure 1 (Left).
> >
> > The same questions remain: if they have different training processes and training data, this *"intuitive comparison of different published results"* weakens the support for the paper's conclusions. A fair comparison is highly valued in the field of computer vision.
> >
> > I strongly recommend that the authors include more equitable experimental results in the paper, as this would greatly enhance its quality. If such changes are not feasible at this stage, I believe my rating is fair and reasonable.

---

> ### Comment · Reviewer_cPqJ · 2024-08-13
>
> Thank you for the detailed response. I appreciate the author's candid acknowledgment of the limitations of the current work. Given that these limitations objectively exist, this discussion will not change my evaluation.
>
> I would like to clarify two points:
> - The 7% I mentioned is intended to highlight the relative improvement potential, calculated as $(85.2 - 84.6) / 84.6 \times 100 = 7 $%.
> - I hope the author is aware that, as shown in Figure 1 (Right), the additional FLOPs required for CRATE-α to match the comparable performance of ViT increases significantly with scale.

---

> ### Author Response · Authors · 2024-08-13
> **Response to Reviewer cPqJ**
>
> Thank you for your valuable and in-depth review, as well as the series of insightful discussions. Your feedback and suggestions will undoubtedly enhance the quality of the paper. We greatly appreciate your input.
>
> Regarding the 'more equitable experimental results', according to the NeurIPS discussion policy, "*The discussion phase is meant to clarify these questions, rather than to provide further comments regarding the reviews.*" Due to limited time and resource constraints, we are not able to provide new experimental results under more equitable settings before the end of the discussion phase. We will conduct new experiments as per your suggestions and incorporate them into our revised version.

---

### Official Review · Reviewer_yemz · 2024-07-13

**Soundness:** 3
**Presentation:** 3
**Contribution:** 2
**Rating:** 5
**Confidence:** 4

**Summary:**

This paper studies the scalability problem of white-box transformer CRATE and proposes CRATE-$\alpha$ to enhance the scaling ability of CRATE. To be specific, the authors propose three strategic but minimal modifications for the CRATE model architecture: Overparameterized sparse coding block, Decoupled dictionary, and Residual connection. Extensive experiments across different datasets and settings demonstrate the effectiveness of the proposed approach.

**Strengths:**

1. It is quite meaningful to study white-box transformers and try to increase their scalability which promises its application in potential usage.

2. Comprehensive evaluation. The proposed method is validated on multiple datasets and tasks which demonstrate the scalability of CRATE-$\alpha$.

3. The presentation is clear. Overall, the paper is well-organized and the method is easy to follow.

**Weaknesses:**

1. Performance gaps with vanilla ViT. As shown in Figure 1, CRATE-$\alpha$ still lags behind vanilla ViT across different scales remarkably which may limit its application in real scenarios. Besides, it is suggested to compare with vanilla ViT in computational costs, number of parameters, and inference speed as well.

2. According to the model configuration, the number of parameters of CRATE-$\alpha$ is almost four times as CRATE and it is strange to consider those as the same scale models. Moreover, how do the proposed new modules contribute to the performance gain of CRATE-$\alpha$? Is it simply because of larger models?

3. Although the authors made lots of efforts in scaling CRATE to CRATE-$\alpha$, they only spent limited space in the paper to discuss the interpretability of the proposed method. This short paragraph may not be enough to justify why the authors are motivated to study the white-box transformers.

**Questions:**

Apart from the questions in weakness above, another question is:

why the performance in dense prediction tasks is so bad?

---

> ### Author Rebuttal · Authors · 2024-08-07
>
> Thank you for your review. Below we attempt to resolve the questions you posed.
>
> >**Q1**: *Performance gaps with vanilla ViT. As shown in Figure 1, CRATE-α still lags behind vanilla ViT across different scales remarkably which may limit its application in real scenarios. Besides, it is suggested to compare with vanilla ViT in computational costs, number of parameters, and inference speed as well.*
>
> **A1**: Thank you for your suggestions on comparison between ViT and CRATE-α. Please refer to our response to '**Q1: Comparison with ViT**' in the common response.
>
> >**Q2**: *According to the model configuration, the number of parameters of CRATE-α is almost four times as CRATE and it is strange to consider those as the same scale models. Moreover, how do the proposed new modules contribute to the performance gain of CRATE-α? Is it simply because of larger models?*
>
> **A2**: Considering the number of parameters, CRATE-α-B/16 (72.3M) is comparable to CRATE-L/16 (77.6M), as shown in table 4 of the Appendix. CRATE-α-B/16 still surpasses CRATE-L/16 by 9.9% (81.2% vs 71.3%) in ImageNet-1K (see Table 1 of the manuscript). Meanwhile, from Table 1 of the manuscript, we also observe that increasing the size of CRATE from base (22.8M) to large (77.6M) results in only a 0.5% improvement, demonstrating diminished returns at the cost of 3.4 times more parameters. In contrast, we found that change from CRATE-B/16 to CRATE-α-B/16 leads to a much more significant improvement. Moreover, we also tried directly scaling CRATE-L to a larger dataset (ImageNet-21K), but the experiment showed it is challenging to train CRATE-L directly due to the instability of the optimization, and its performance is even worse than just using ImageNet-1K.
>
> Therefore, combining these observations, we can find that the design of CRATE-α indeed leads to improved performance than CRATE, which is not simply due to the number of parameters being larger. Without a carefully designed architecture, naively increasing parameters might not lead to improved performance. Thank you for your question, and we will add the above discussion to our revised version.
>
> >**Q3**: *Although the authors made lots of efforts in scaling CRATE to CRATE-α, they only spent limited space in the paper to discuss the interpretability of the proposed method. This short paragraph may not be enough to justify why the authors are motivated to study the white-box transformers.*
>
> **A3**: Thank you for your suggestion on the interpretability part of the paper presentation. *The main focus and motivation* of this paper is to investigate whether it is possible to scale up white-box deep learning models to achieve competitive performance as black-box ones such as ViTs, which is a *long-standing problem for white-box deep learning models* [GL2010, ZTA+2019, DLZ+2022, YBP+2023]. Our work *provides an affirmative answer* to this question.
>
> Meanwhile, the previous work [YBP+2023, PBW+2024] has already explored the mathematical interpretability of white-box transformers. Our newly designed white-box model CRATE-α follows the white-box design principles as previous work, and the operators and architectures are derived by optimizing the sparse coding rate reduction Eq. (1) in the manuscript. Therefore, the mathematically interpretability [YBP+2023, PBW+2024] developed in previous work is inherited by this work. It would be highly interesting to provide a more comprehensive study on the interpretability of  CRATE-α, such exploration is out-of-scope for the present work and left for future work.
>
> Per your suggestions, we have conducted more experiments on the zero-shot segmentation (following the same setup in Section 4.4), including segmentation visualization on more images as shown in Figure 5. We will add these new experimental results to the appendix of our revision.
>
> >**Q4**: *Why the performance in dense prediction tasks is so bad?*
>
> **A4**: The reported results in Table 3 of the manuscript are evaluated **under the zero-shot segmentation setting**. For the results in Table 3, we were using MaskCut [WGY+2023], a self-supervised segmentation method, to study the interpretability of different models (ViT, CRATE, CRATE-α). In particular, we directly extracted features using the learned models and applied MaskCut to perform segmentation. We did not perform supervised training on segmentation tasks. Therefore, the segmentation results are not as competitive as the standard ones which are typically trained with segmentation labels. Thank you for your suggestion. We will add a paragraph and highlight this difference in our revised version.
>
> We again thank you for your review, and hope we have provided satisfactory responses to your questions. Please let us know if you have further questions or comments.
>
> [WGY+2023] Cut and learn for unsupervised object detection and instance segmentation. Xudong Wang, Rohit Girdhar, Stella X Yu, et al. CVPR 2023.
>
> [GL2010] Learning fast approximations of sparse coding. Karol Gregor and Yann LeCun. ICML 2010.
>
> [ZTA+2019] Deep Network Classification by Scattering and Homotopy Dictionary Learning. John Zarka, Louis Thiry, Tomás Angles, et al. arXiv:1910.03561. 2019.
>
> [DLZ+2022] Revisiting sparse convolutional model for visual recognition. Xili Dai, Mingyang Li, Pengyuan Zhai, et al. NeurIPS 2022.
>
> [YBP+2023] White-Box Transformers via Sparse Rate Reduction. Yaodong Yu, Sam Buchanan, Druv Pai, et al. NeurIPS 2023.
>
> [PBW+2024] Masked Completion via Structured Diffusion with White-Box Transformers. Druv Pai, Sam Buchanan, Ziyang Wu, et al. ICLR 2024.

---

> > ### Comment · Reviewer_yemz · 2024-08-13
> > **Post-rebuttal comment**
> >
> > The reviewer appreciates the detailed response provided by the authors. Most of the concerns are addressed and the rating is upgraded accordingly. One suggestion that the authors could consider is adding the experiments of supervised training on segmentation tasks which is important to examine the dense prediction ability of white-box transformers.

---

> ### Author Response · Authors · 2024-08-13
> **Response to Reviewer yemz**
>
> Thank you again for thoroughly reviewing our manuscript and response. We are grateful for your valuable feedback on our work, which will no doubt improve it.  We will add the results of supervised segmentation tasks to the main body of the revised version.

---

### Official Review · Reviewer_8F4z · 2024-07-13

**Soundness:** 2
**Presentation:** 3
**Contribution:** 3
**Rating:** 6
**Confidence:** 3

**Summary:**

This paper aims to train CRATE at a large scale for vision tasks. The contribution includes an architecture modification to the sparse coding block and a light training recipe. The new model, called CRATE-alpha, shows large improvements compared with the previous CRATE model. The experiments also show promising results on unsupervised object segmentation.

**Strengths:**

- The paper presents a careful study to enhance the performance of CRATE. The paper introduces key modifications to the existing CRATE, including the sparse coding block, decoupled dictionary, and residual connection.
- The paper investigates its scaling behavior and shows promising improvements of the newly introduced CRATE-alpha.
- The paper presents in-depth experiments, such as the scaling analysis on ImageNet. The paper also shows improvements for semantic interpretability.
- The figures and model architecture are well-illustrated.

**Weaknesses:**

Overall I find the paper is well-presented and solid. Below are my minor concerns for this paper:
- The paper is highly centered on improving CRATE. Most of the findings might not be transferable to other models. This may limit its impact to the general audience in NuerIPS community.
- It would be interesting to further understand its potential downstream applications (not only vision but also language data)

**Questions:**

see weakness

---

> ### Author Rebuttal · Authors · 2024-08-07
>
> Thank you for your review. Below we attempt to resolve the questions you posed.
>
> >**Q1**: *The paper is highly centered on improving CRATE. Most of the findings might not be transferable to other models. This may limit its impact to the general audience in NuerIPS community.*
>
> **A1**: We agree that some of our findings mainly apply to white-box models (e.g., white-box transformers like CRATE [YBP+2023]). However, there is growing interest in developing white-box transformers for better interpretability and transparency across a wide range of tasks and domains, including image segmentation [YBP+2023], self-supervised masked autoencoders [PBW+2024], and integrated sensing and communications [ZL2024], etc. We believe that our findings and insights could be helpful for developing white-box transformers for a wide range of applications and tasks. Moreover, our results on the scalability of white-box transformers could also shed light on scaling up a broader class of white-box deep neural networks, such as white-box ISTA networks and their variants [GL2010, SNT2018, CLW+2018, ZTA+2019, DLZ+2022] designed via unrolled optimization. In summary, we believe that the findings and insights of this work could benefit a broad audience in the NeurIPS community interested in building more interpretable and performant deep learning models.
>
> Thank you for your suggestion on discussing the impact of this work, and we add the above discussion to our revision.
>
>
> >**Q2**: *It would be interesting to further understand its potential downstream applications (not only vision but also language data)*
>
> **A2**: Thank you for your suggestion. Please refer to our response to '**Q2: Additional experimental results on real-world downstream tasks**'  and '**Q3: Performance of CRATE-α on NLP task**' in the common response.
>
> We again thank you for your review, and hope we have provided satisfactory responses to your questions. Please let us know if you have further questions or comments.
>
> [YBP+2023] White-Box Transformers via Sparse Rate Reduction. Yaodong Yu, Sam Buchanan, Druv Pai, et al. NeurIPS 2023.
>
> [PBW+2024] Masked Completion via Structured Diffusion with White-Box Transformers. Druv Pai, Sam Buchanan, Ziyang Wu, et al. ICLR 2024.
>
> [ZL2024] White-Box 3D-OMP-Transformer for ISAC. Bowen Zhang, Geoffrey Ye Li. arXiv:2407.02251.
>
> [GL2010] Learning fast approximations of sparse coding. Karol Gregor and Yann LeCun. ICML 2010.
>
> [SNT2018] Supervised deep sparse coding networks. Xiaoxia Sun, Nasser M Nasrabadi, Trac D Tran. ICIP 2018.
>
> [CLW+2018] Theoretical Linear Convergence of Unfolded ISTA and its Practical Weights and Thresholds. Xiaohan Chen, Jialin Liu, Zhangyang Wang, et al. NeurIPS 2018.
>
> [ZTA+2019] Deep Network Classification by Scattering and Homotopy Dictionary Learning. John Zarka, Louis Thiry, Tomás Angles, et al. arXiv:1910.03561. 2019.
>
> [DLZ+2022] Revisiting sparse convolutional model for visual recognition. Xili Dai, Mingyang Li, Pengyuan Zhai, et al. NeurIPS 2022.

---

> > ### Comment · Reviewer_8F4z · 2024-08-10
> > **comments**
> >
> > Thank you. I have no further comment at this point.

---

### Author Rebuttal · Authors · 2024-08-07

### **Common response to all reviewers**:

We thank all reviewers for their insightful feedback. We are especially encouraged by their recognition of:

- The novelty and impact of our central ideas (` Reviewer YH3U `: “The paper presents a novel architecture, CRATE-α, …, enhancing scalability without compromising interpretability.” ` Reviewer cPqJ `: “successfully combining them with a white-box Transformer is a noteworthy achievement. The integration not only works effectively but also yields commendable results.”)
- The benefits of the new white-box architecture we have proposed (` Reviewer 8F4z `: “shows promising improvements of the newly introduced CRATE-alpha.”; ` Reviewer yemz `: “Comprehensive evaluation.”; ` Reviewer cPqJ `: “The proposed models demonstrate significant improvements compared to the previous generation of CRATE models.”; ` Reviewer YH3U `: “The authors provide a wealth of empirical evidence supporting the effectiveness of CRATE-α”)
- The quality of the exposition (` Reviewer 8F4z `: “The figures and model architecture are well-illustrated.”; ` Reviewer yemz `: “the paper is well-organized and the method is easy to follow.”)
- The motivation of our work (` Reviewer yemz `: “It is quite meaningful to study white-box transformers and try to increase their scalability …”)

In the remainder of this message, we address certain questions raised by the reviewers.

> ### **Q1: Comparison with ViT**

**A1**: We agree that the proposed white-box CRATE-α is not significantly outperforming ViT on classification tasks. However, based on the newly designed dictionary learning block and training recipe, we were able to significantly improve over the vanilla white-box CRATE and nearly closed the gap between white-box transformers and ViTs, especially when the model size becomes larger.

**Performance comparison.** In Figure 1 (right), ViT was fine-tuned at 384x384 resolution, while CRATE was at 224x224. To more accurately compare CRATE-α and ViT with larger model sizes, we conducted experiments on CRATE-α-L/16 with an image resolution of 336, nearly matching the setup of ViT-L/16. Both models used a similar amount of FLOPs: 210G for CRATE-α-L/16 compared to 191G for ViT-L/16. The throughput, or images processed per second, was also comparable at 35.53 for our model versus 35.56 for ViT-L/16. The accuracy of CRATE-α-L/16 reached 84.6%, closely approaching ViT’s 85.2% under similar conditions. Meanwhile, combining the trend from Figure 1 (right), this narrowing performance gap from *Base* to *Large* model size suggests that CRATE-α can nearly matche ViT’s performance in large-scale settings. Besides, CRATE-α inherits the mathematical interpretability of the white-box models and can also achieve much better semantic interpretability evaluated by zero-shot segmentation.

**Efficiency comparison.** We would like to thank ` Reviewer yemz ` for suggestions on comparison with ViT in computational costs, number of parameters, and inference speed. These comparisons are summarized in the Table 1 of our rebuttal pdf, where CRATE-α matches the efficiency of ViT while achieving similar accuracy. With the same layers/embedding dimension, CRATE-α has fewer parameters than ViT, and the FLOPs/Throughput of CRATE-α is slightly higher than ViT.

We will add the above discussion and new experimental results to our revised version.

> ### **Q2: Additional experimental results on real-world downstream tasks**

**A2:** We have conducted new experiments on supervised segmentation tasks and fine-tuning on additional downstream datasets.

**Segmentation.** We compare the performance of CRATE and CRATE-α for the segmentation task on ADE20K dataset to study the benefits of the newly designed architecture CRATE-α compared to CRATE. Following the setup of [RWZ+2023], we compared them in segmentation tasks, focusing on direct comparisons due to time and resource constraints, without extensive parameter tuning. Results demonstrate that CRATE-α consistently outperforms CRATE across all key metrics, with both models pre-trained on IN21K. These findings indicate significant performance gains in vision tasks beyond classification. We will include these results in our revised version.

| Model        | Scope  | mIoU  | mAcc  | aAcc  |
| ------------ | ------ | ----- | ----- | ----- |
| CRATE-α-B/32 | global | 35.35 | 45.28 | 77.63 |
| CRATE-B/32   | global | 30.28 | 39.29 | 75.21 |

**Fine-tuning classification datasets.** We included additional experimental results on evaluations on downstream datasets for CRATE-α. From the table 2 of our rebuttal pdf, we find that CRATE-α consistently outperforms CRATE (both models are pre-trained on IN21K), and CRATE-α achieves improved performance when model size increases.

We will add the above new results to our revised version.

> ### **Q3: Performance of CRATE-α on NLP task**

**A3**: Thanks to ` Reviewers 8F4z ` and ` Reviewer YH3U ` for suggesting we explore CRATE-α in language tasks. We conducted new experiments for CRATE-α with autoregressive training on OpenWebText, following the setup in [Karpathy2022]. We compare CRATE-α models (57M and 120M) with CRATE and GPT-2, where the result of CRATE is from [YBP+2023]. Table 3 in our rebuttal pdf shows that CRATE-α significantly improves over CRATE in language modeling. Due to limited time and resource constraints, compared to the total number of iterations (600K) used in CRATE, we only finished 80% of the total iterations for CRATE-α-small and 55% for CRATE-α-base. CRATE-α still showed notable improvements. We will add these new results to our revised version.

[Karpathy2022] NanoGPT. https://github.com/karpathy/nanoGPT. Andrej Karpathy

[YBP+2023] White-Box Transformers via Sparse Rate Reduction: Compression Is All There Is? Yaodong Yu, Sam Buchanan, Druv Pai, et al. arXiv:2311.13110. 2023.

[RWZ+2023] TinyMIM: An Empirical Study of Distilling MIM Pre-Trained Models. Sucheng Ren, Fangyun Wei, Zheng Zhang, et al. CVPR 2023.

---

### Decision · Program_Chairs · 2024-09-25

**Decision:**

Accept (poster)

**Comment:**

After rebuttal and discussion, all reviewers agree in their recommendation to accept this paper. The AC and SAC see no reason to overturn this recommendation, and hence decide to accept this paper. Congratulations.